# IntLoRA: Integral Low-rank Adaptation of Quantized Diffusion Models

## Abstract

Fine-tuning large-scale text-to-image diffusion models for various downstream tasks has yielded impressive results. However, the heavy computational burdens of tuning large models prevent personal customization. Recent advances have attempted to employ parameter-efficient fine-tuning (PEFT) techniques to adapt the floating-point (*e.g.*, FP16) or quantized pre-trained weights. Nonetheless, the adaptation parameters in existing works are still restricted to FP16 arithmetic, hindering hardware-friendly acceleration. In this work, we propose IntLoRA, to further push the efficiency limits by using integer type (*e.g.*, INT4) low-rank parameters to adapt the quantized diffusion models. By working in the integer arithmetic, our IntLoRA offers three key advantages: (i) for fine-tuning, the pre-trained weights are quantized, reducing memory usage; (ii) for storage, both pre-trained and low-rank weights are in INT4 which consumes less disk space; (iii) for inference, IntLoRA weights can be naturally merged into quantized pre-trained weights through efficient integer multiplication or bit-shifting, eliminating additional post-training quantization. Extensive experiments demonstrate that IntLoRA can achieve performance on par with or even superior to the vanilla LoRA, accompanied by significant efficiency improvements.

## 1 Introduction

Recently, large-scale text-to-image (T2I) diffusion models (Ramesh et al., 2022; Rombach et al., 2022; Saharia et al., 2022; Podell et al., 2023) have shown promising capabilities for image generation. Taking advantage of the strong generative prior of pre-trained parameters, a range of downstream adaptation applications have emerged, including subject-driven generation (Ruiz et al., 2023), style-customized generation (Sohn et al., 2023), and controllable generation (Zhang et al., 2023). However, fully fine-tuning these large models for personalized customization poses challenges on consumer-level GPUs, as well as the costs of storing the weight for each downstream task.

To facilitate efficient adaptation, recent advances (Qiu et al., 2023; Liu et al., 2023b) have introduced parameter efficient fine-tuning (PEFT) (Hu et al., 2021; Houlsby et al., 2019; Jia et al., 2022) that focuses on fine-tuning a limited number of parameters for downstream tasks. Despite the reduction in trainable parameters, current PEFT methods predominantly work on floating-point (*e.g.*, FP16) arithmetic, which can be inefficient for practical applications. For instance, only loading the FLUX.1-dev (BlackForestLabs, 2024) can consume over 23GB GPU memory, let alone subsequent fine-tuning. Additionally, it also costs huge disk space to store per-task fine-tuned models. Although some PEFT methods propose weight merging to mitigate the overhead associated with additional adapters, they still fall short in accelerated inference.

On the other hand, neural network quantization (Nagel et al., 2021; 2020; Esser et al., 2019), which can transform the trained FP16 parameters into low-bit integer (*e.g.*, INT4) representations, is a prominent technique for accelerating deep learning models. Thus, integrating PEFT with quantization techniques holds promise for enhancing the efficiency of downstream adaptations. To this end, some pioneer works (Dettmers et al., 2024; Qin et al., 2024) attempt downstream adaptation by directly fine-tuning the quantized INT4 pre-trained weights with FP16 PEFT parameters. However, it remains an open challenge for quantization-aware adaptation. Specifically, since the adaptation weights are still in FP16, it is inevitable to convert the quantized pre-trained weights back to FP16 for arithmetic consistency for subsequent weight merge. As a result, it necessitates additional post-

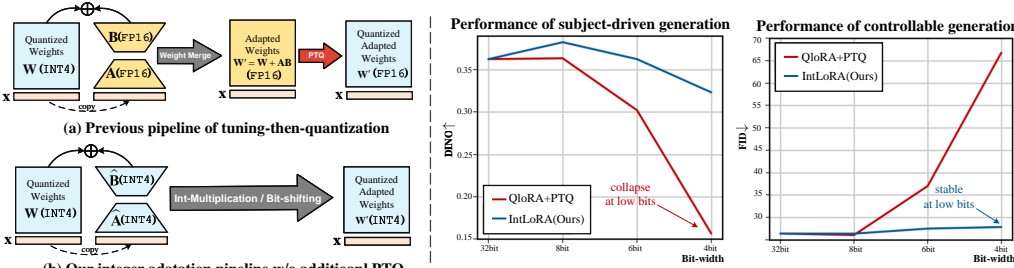

Figure 1: **Left:** (a) The arithmetic inconsistency between the pre-trained and adaptation weights results in the merged weights still in `FP16`. Consequently, additional PTQ is needed for low-bit inference. (b) Our approach allows PEFT to operate directly on `INT4` arithmetic, ensuring the merged weights seamlessly in `INT4` format and streamlining the whole process. **Right:** The utilization of PTQ on the downstream adapted weights leads to severe performance degradation under low bit-width quantization, which makes the tuning-then-quantization paradigm less general.

training quantization (PTQ) to quantize the merged weights for practical employment, which is pipeline-complicated and incurs a performance drop when the bit-width is low (see Fig. 1).

To some extent, the primary technical bottleneck in bridging PEFT and quantization arises from the *arithmetic inconsistency* between the quantized pre-trained weights and the adaptation weights. Specifically, the `FP16` adaptation weights necessities the conversion of the quantized pre-training weights from `INT4` to `FP16` for the subsequent weight fusion. To address this inconsistency, a potential solution is to also transfer the adaptation weights to integer arithmetic. In this way, all weights during fine-tuning are quantized, allowing to store only `INT4` weights, as well as convenient weight merging during inference. Despite these promising properties, it is non-trivial to accurately quantize the adaptation weights for satisfactory performance. For example, while zero initializing low-rank weights are advantageous for fine-tuning (Hu et al., 2021), it poses quantization challenges due to substantial quantization errors from small values. Furthermore, the additive form of the original LoRA forces the pre-trained and adaptation weights to share the same quantizer for subsequent weight merging, which restricts the available parameter space for adaptation.

To address the above challenges, we propose IntLoRA, which employs integral low-rank parameters to adapt the quantized diffusion models. In detail, we introduce the Adaptation-Quantization Separation (AQS) technique, which employs a task-agnostic auxiliary matrix to enable quantization-friendly low-rank parameters without disrupting the gradient trajectory of the original LoRA. Additionally, we present the Multiplicative Low-rank Adaptation (MLA), which reformulates the mathematical structure of LoRA from addition to multiplication. This remains mathematically equivalent to the original but allows for independent optimization of adaptation weights, eliminating the need to share the same quantizer as the pre-trained weights. Furthermore, we develop the Variance Matching Control (VMC) mechanism that aligns variances of the pre-trained and auxiliary matrices, controlling the adaptation distribution for more efficient log2-quantization. For implementation, we provide two versions of IntLoRA, *i.e.*, IntLoRA$_{\text{MUL}}$, and IntLoRA$_{\text{SHIFT}}$. The IntLoRA$_{\text{MUL}}$ learns quantized low-rank parameters and can be seamlessly merged with quantized pre-trained weight through integer multiplication, while IntLoRA$_{\text{SHIFT}}$ introduces log2-quantization and operates by bit-shifting the quantized pre-trained weights for downstream adaptation. We evaluate our IntLoRA on prevalent T2I adaptation tasks, including subject-driven generation (Ruiz et al., 2023), style-customized generation (Sohn et al., 2023) and controllable generation (Zhang et al., 2023). Extensive experiments demonstrate that IntLoRA represents a novel diffusion fine-tuning paradigm with impressive efficiency and favorable performance.

The contribution of this work can be summarized as follows: **(i)** we introduce IntLoRA, which achieves integer PEFT to address the arithmetic inconsistency, thereby advancing the efficiency of diffusion model adaptations; **(ii)** we propose the adaptation-quantization separation to facilitate quantization-friendly pre-trained weights, and further develop the multiplicative low-rank adaptation for independent quantizers, complemented by variance matching control for effective distribution manipulation; **(iii)** our IntLoRA enables the adaptation of quantized diffusion models through hardware-friendly integer multiplication or bit-shifting, resulting in significant efficiency gains in fine-tuning, storage, and inference. Extensive experiments validate the superiority of our method.

## 2 RELATED WORK

**Parameter-efficient fine-tuning of diffusion models.** In order to reduce the fine-tuning cost of large models, parameter-efficient fine-tuning (PEFT) has recently gained great interests (Lian et al., 2022; Chavan et al., 2023; Li & Liang, 2021; He et al., 2021; Jie & Deng, 2023). For example, some pioneering work (Zaken et al., 2021) proposes to fine-tune only a fraction of the pre-trained weights. To achieve better performance, prompt-based methods (Jia et al., 2022) append learnable prompts to modify the input space. Adapter-based methods (Houlsby et al., 2019; Chen et al., 2022) employ additional bottleneck structures as bypass branches for adaptation. LoRA (Hu et al., 2021), which adopts low-rank matrices to learn weight updates for downstream tasks, has become a popular pipeline for diffusion model adaptation. Moreover, OFT (Qiu et al., 2023) uses orthogonal constraints to preserve the pairwise angles between neuron vectors to maintain the hypersphere energy. COFT (Liu et al., 2023b) further uses butterfly factorization to formulate denser orthogonal matrices using fewer parameters. In this work, we mainly focus on LoRA since it has been widely applied and can be merged into pre-trained weights without increasing the inference cost.

**Network quantization of diffusion models.** Quantization (Nagel et al., 2021) is an effective technique to reduce model storage and inference costs, and can be categorized into quantization-aware training (QAT) (Jacob et al., 2018; Li et al., 2024; 2022; Xu et al., 2023a) and post-training quantization (PTQ) (Wang et al., 2023a; Nahshan et al., 2021; Li et al., 2021; Wei et al., 2022; Liu et al., 2023a; Huang et al., 2024a). In the context of diffusion model quantization, existing works mainly focus on PTQ because of the significant overhead of retraining diffusion models. For example, PTQ4DM (Shang et al., 2023) makes the first attempt to quantize diffusion models to 8 bits. After that, Q-Diffusion (Li et al., 2023) further achieves improved performance and lower bit-width. In addition, PTQD (He et al., 2024) eliminates quantization noise through correlated and residual noise correction. EfficientDM (He et al., 2023) introduces LoRA to fine-tune the pre-trained model to allow comparable performance with QAT. TFMQ-DM (Huang et al., 2024b) observes the impact of quantization errors on the time-step representation and proposes to quantize the time-embedding layer individually for better performance. However, the network quantization only enables efficient inference at low bit-widths on the original task, failing to handle various downstream tasks.

**Joint adaptation and quantization.** The work aims to achieve both parameter-efficient adaptations as well as storage and inference-efficient quantization, to further push the efficiency limits of diffusion model fine-tuning. However, this also poses additional challenges, such as back-propagating gradients in quantized values and optimizing the learnable parameters with quantization restrains. Existing related works have explored mainly LLMs, but it is still far from a satisfactory solution. Specifically, QLoRA (Dettmers et al., 2024) proposes to quantize the LLMs before fine-tuning the LLMs with LoRA. Despite the reduced GPU usage during training due to the import of only the quantized model, QLoRA does not maintain quantized at inference since the quantized weights need to be converted to `FP16` again so that to add the LoRA weights. To solve this problem, QA-LoRA (Xu et al., 2023b) develops a group-wise quantization through sharing parameters across channels, but at the cost of impairing the adaptation ability as well as the need to store `FP16` LoRA weights. IR-QLoRA (Qin et al., 2024) analyzes the entropy loss of quantization from an information theory view, but it also needs to convert the quantized weights back to `FP16` during inference.

## 3 PRELIMINARY

The main idea of low-rank adaptation (LoRA) (Hu et al., 2021) comes from the fact that the learned incremental matrix during downstream adaptation usually possesses low-rank properties. To this end, LoRA introduces a low-rank matrix $\Delta\mathbf{W}$ to learn the weight increments for adapting the pre-trained weights $\mathbf{W} \in \mathbb{R}^{C_{out} \times C_{in}}$ to downstream tasks. In implementation, the $\Delta\mathbf{W}$ is formulated as the matrix multiplication of two low-rank matrices $\mathbf{A} \in \mathbb{R}^{C_{out} \times d}$ and $\mathbf{B} \in \mathbb{R}^{d \times C_{in}}$, where the inner dimension $d$ is the pre-defined rank. During fine-tuning, the pre-trained weight $\mathbf{W}$ is frozen and only $\mathbf{A}, \mathbf{B}$ is trainable. Since $d \ll \min\{C_{in}, C_{out}\}$, the number of trainable parameters can be very small compared to full fine-tuning. The output during the downstream fine-tuning is calculated as $\mathbf{y} = \mathbf{W}\mathbf{x} + \lambda \cdot (\mathbf{A}\mathbf{B})\mathbf{x}$, where $\lambda$ is the LoRA scale to adjust the control strength. During inference, the task-specific $\mathbf{A}\mathbf{B}$ can be naturally merged into the pre-trained weights, *i.e.*, $\mathbf{W}' = \mathbf{W} + \lambda \cdot \mathbf{A}\mathbf{B}$, without increasing additional computational cost.

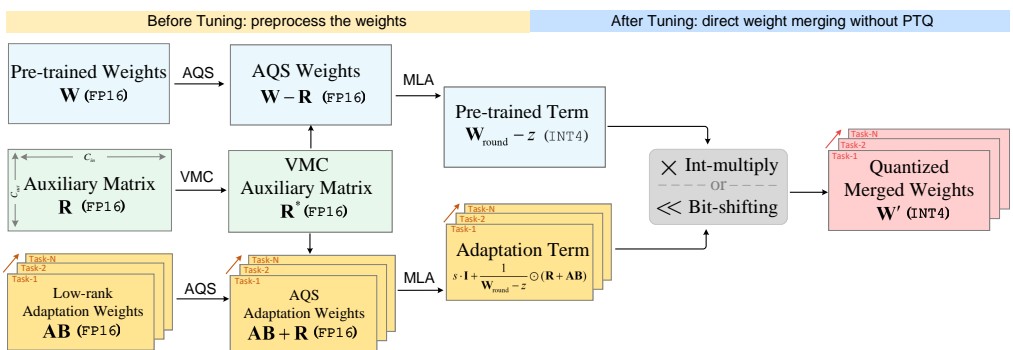

Figure 2: **Before tuning**, we propose the Adaptation Quantization Separation (AQS) to incorporate auxiliary matrix into pre-trained weights and low-rank weights to obtain zero initialized but quantization-friendly parameters. Then, the Multiplicative Low-rank Adaptation (MLA) is used to reformulate additive LoRA into the product of the "pre-training term" and the "adaptation term". At last, we introduce the Variance Matching Control (VMC) to adjust the distribution of adaptation term by modulating the auxiliary matrix. **After tuning**, we use hardware-friendly integer multiplication or bit shifting to directly generate quantized downstream weights without additional PTQ.

Even though the PEFT techniques can alleviate training costs, they still struggle to reduce the inference latency. To allow for accelerated inference, network quantization is a common practice that converts the floating-pointing weights to hardware-efficient low-bit integers. Formally, given a tensor $\mathbf{X}$, the target bit-width $b$, the quantization process can be defined as:

$$\hat{\mathbf{X}} = s \cdot (\text{clip}(\lfloor \frac{\mathbf{X}}{s} \rceil + z, 0, 2^b - 1) - z) \triangleq s \cdot (\mathbf{X}_{\text{round}} - z), \tag{1}$$

where $\lfloor \cdot \rceil$ is the round function, $s = \frac{\max(\mathbf{X}) - \min(\mathbf{X})}{2^b - 1}$ is the scaling factor, and $z = -\lfloor \frac{\min(\mathbf{X})}{s} \rceil$ is the zero-point. In short, PEFT and quantization can facilitate efficient training and inference, respectively, and thus integrating both into one system holds great potential for further acceleration.

## 4 METHODOLOGY

### 4.1 BRIDGING EFFICIENT-ADAPTATION AND QUANTIZATION

Although parameter-efficient adaptation and network quantization each contribute to improved fine-tuning and inference efficiency, it remains an unexplored challenge to effectively combine them. Existing work such as QLoRA (Dettmers et al., 2024) employs the low-rank $\mathbf{AB}$ to fine-tune quantized weights $\hat{\mathbf{W}}$ to reduce the memory footprint of loading a full-precision model. After training, the merging between low-rank FP16 parameters and quantized NF4 weights causes the adapted weights $\mathbf{W}'$ to revert to FP16 again. Consequently, additional PTQ on $\mathbf{W}'$ is needed for inference efficiency. When the bit width is low, *e.g.*, 4-bit, this PTQ can significantly degrade performance.

In this work, we attribute the main difficulty in effectively marrying PEFT and quantization to the arithmetic inconsistency. Specifically, the FP16 $\mathbf{AB}$ forces the quantized $\hat{\mathbf{W}}$ to revert to FP16 during weight merging. To bridge the arithmetic gap, a feasible solution is to convert the low-rank parameters into integer type to obtain $\hat{\mathbf{AB}}$. In this way, the same typed $\hat{\mathbf{W}}$ and $\hat{\mathbf{AB}}$ can be seamlessly merged during inference without additional PTQ. However, several technical challenges arise when performing PEFT on integer arithmetic. First, the $\mathbf{AB}$ in the original LoRA is zero-initialized to preserve the knowledge of the pre-trained weights. Although helpful for fine-tuning, this initialization complicates the quantization process. For instance, the all-zero distribution requires a separately designed quantizer at the beginning of tuning, since the scaling factor $s = 0$ leads to an infinite $\frac{\mathbf{X}}{s}$ in Eq. (1). Second, the vanilla LoRA merges the FP16 $\mathbf{AB}$ and $\mathbf{W}$ using addition. When both $\hat{\mathbf{AB}}$ and $\mathbf{W}$ are quantized, it is essential to ensure that they share identical quantization parameters to enable PTQ-free weight merging. This requirement leads to constrained parameter space, thus limiting the adaptation ability.

### 4.2 INTEGRAL LOW-RANK ADAPTATION

To address the above challenges, we propose the integral low-rank adaption, dubbed IntLoRA, which enables PEFT in integer arithmetic. The overall pipeline is shown in Fig. 2.

**Adaptation-quantization separation.** The vanilla LoRA adopts zero initialization on $\mathbf{AB}$ to ensure the behavior of the model is similar to the pre-trained one at the beginning of training (Hu et al., 2021). Although this initialization can improve performance, the all-zero distribution is not quantization-friendly, as validated in Sec. 4.1 and Sec. 5.3. To allow accurate quantization while maintaining the correct gradient, we propose the Adaptation-Quantization-Separation (AQS) strategy. The key observation is that the adaptation requires gradients from zero-initialized weights while the quantization does not. Therefore, we can split the non-zero initialized adaptation weights into the gradient-aware zero part and the gradient-free nonzero part. Formally, let $\mathbf{R}$ be the non-zero auxiliary matrix, $\mathcal{Q}$ be the quant-dequant operator, then our AQS can be formulated as:

$$\mathbf{W}' = \mathcal{Q}[\mathbf{W} - \mathrm{sg}(\mathbf{R})] + \mathrm{sg}(\mathbf{R}) + \mathbf{AB}, \tag{2}$$

where $\mathrm{sg}(\cdot)$ denotes the stop gradient operation. Thanks to the AQS, the $\mathbf{AB}$ can be zero-initialized for the same gradient as the original LoRA, while $\mathrm{sg}(\mathbf{R}) + \mathbf{AB}$ facilitate subsequent quantization by specifically designing the auxiliary matrix $\mathbf{R}$ as discussed in Sec. 5.3. In the following part, we will ignore the $\mathrm{sg}(\cdot)$ notation for clarity.

**Multiplicative low-rank adaptation.** The vanilla LoRA employs additive form $\mathbf{W} + \mathbf{AB}$ for weight merge. However, it is difficult to seamlessly fuse the quantized $\hat{\mathbf{W}}$ and $\hat{\mathbf{AB}}$ when they are quantized by independent quantizers. To this end, we propose Multiplicative Low-rank Adaptation (MLA) to rewrite the form of the original LoRA into a quantization-friendly multiplication form. Specifically, denote the quant-dequant results as $\mathcal{Q}(\mathbf{W} - \mathbf{R}) = s \cdot (\mathbf{W}_{\mathrm{round}} - z)$, then the MLA can be derived as follows:

$$\begin{aligned}
\mathbf{W}' &= \mathcal{Q}(\mathbf{W} - \mathbf{R}) + \mathbf{R} + \mathbf{AB} \\
&= s \cdot (\mathbf{W}_{\mathrm{round}} - z) + \mathbf{R} + \mathbf{AB} \\
&= [s \cdot \mathbf{I} + \frac{1}{\mathbf{W}_{\mathrm{round}} - z} \odot (\mathbf{R} + \mathbf{AB})] \odot (\mathbf{W}_{\mathrm{round}} - z),
\end{aligned} \tag{3}$$

where the task-specific adaptation term is trainable and will be quantized, and the pre-trained term is already in integer type and is shared across tasks. $\mathbf{I}$ is an all-one matrix. The operator $\odot$ denotes the Hadamard product of two matrices. The proposed MLA is mathematically equivalent to its additive counterpart, while is more quantization-friendly since it avoids the shared quantizer of pre-trained and adaptation weights. It is noteworthy that the adaptation term is still in `FP16`, and we will detail its quantization strategies in Sec. 4.3.

**Variance matching control.** One opportunity brought from the multiplicative form in Eq. (3) is that we can apply the log2-quantization on the adaptation term, thus allowing hardware-efficient bit-shifting on the pre-trained term for adaptation. However, log2-quantization is usually more difficult than common uniform quantization (Nagel et al., 2021) and requires appropriate distribution properties, *e.g.*, most values concentrated around zero to allow for the utilization of as many quantization bins as possible on the logarithmic scale. Here, we revisit the adaptation term in Eq. (3) aiming to find useful mathematical insights. Given the $\mathbf{AB}$ is orders of magnitude smaller than $\mathbf{R}$ (the justification is shown in Appendix L), we approximate the adaptation term in Eq. (3) by removing $\mathbf{AB}$ from it, namely,

$$s \cdot \mathbf{I} + \frac{\mathbf{R}}{\mathbf{W}_{\mathrm{round}} - z} = s \cdot \mathbf{I} + \frac{s \cdot \mathbf{R}}{s \cdot (\mathbf{W}_{\mathrm{round}} - z)} \approx s \cdot \mathbf{I} + \frac{s \cdot \mathbf{R}}{\mathbf{W} - \mathbf{R}} = \frac{s \cdot \mathbf{W}}{\mathbf{W} - \mathbf{R}}. \tag{4}$$

From this derivation, it follows that the auxiliary matrix $\mathbf{R}$ is crucial for controlling the distribution of the adaptation term. However, there exists a dilemma in choosing an appropriate distribution for $\mathbf{R}$. On the one hand, it is desirable for the values in $\mathbf{R}$ to be larger (which are controlled by the variance of $\mathbf{R}$) as it leads to a zero-mean adaptation term, namely,

$$\mathbb{E}\left[\lim_{\sigma_{\mathbf{R}} \to \infty} (s \cdot \mathbf{I} + \frac{\mathbf{R}}{\mathbf{W}_{\mathrm{round}} - z})\right] = \mathbb{E}\left[\lim_{\sigma_{\mathbf{R}} \to \infty} \frac{s \cdot \mathbf{W}}{\mathbf{W} - \mathbf{R}}\right] = 0, \tag{5}$$

where $\sigma_{\mathbf{R}}$ is the standard deviation of $\mathbf{R}$. On the other hand, too large values in $\mathbf{R}$ can also make the term-to-be-quantized $\mathbf{W} - \mathbf{R}$ uncorrelated to the useful $\mathbf{W}$, namely,

$$\lim_{\sigma_{\mathbf{R}} \to \infty} \rho(\mathbf{W} - \mathbf{R}, \mathbf{W}) = \lim_{\sigma_R \to \infty} \frac{\sigma_{\mathbf{W}}}{\sqrt{\sigma_{\mathbf{W}}^2 + \sigma_{\mathbf{R}}^2}} = 0. \tag{6}$$

This low correlation coefficient $\rho$ results in significantly lossy reconstruction of $\mathbf{W}$ from the $\mathbf{W} - \mathbf{R}$ using the estimator $\mathbf{W} \approx \mathcal{Q}(\mathbf{W} - \mathbf{R}) + \mathbf{R}$. Thus, the sampling values of $\mathbf{R}$ cannot be too large. We also give the visualization of this choice dilemma in Fig. 6. Due to the contradictory aspects of the desired properties, it becomes non-trivial to select a proper variance for $\mathbf{R}$. To this end, we propose the Variance Matching Control (VMC) mechanism to adjust the distribution of $\mathbf{R}$. Specifically, we first multiply $\mathbf{R}$ by the variance ratio $r = \frac{\sigma_{\mathbf{W}}}{\sigma_{\mathbf{R}}} \in \mathbb{R}^{C_{out}}$ for rough alignment from $\mathbf{R}$ to the scale of $\mathbf{W}$. After that, we further introduce a scalar $\alpha$ as an exponent of $r$, i.e., $r^\alpha$, to fine-grain the search for the optimal $\mathbf{R}^*$. As a result, the variance-matched auxiliary matrix can be denoted as $\mathbf{R}^* = r^\alpha \cdot \mathbf{R}$, and we can use this to obtain the distribution suitable for log2-quantization. Since $r^\alpha$ can be shared across tasks, it is only of negligible cost. In addition to the $\sigma_{\mathbf{R}}$, we observe the distribution shape of $\mathbf{R}$ also has an effect on performance, and we give a detailed discussion in Sec. 5.3. It should be noted that the $\mathbf{R}$ can be on-line generated during fine-tuning using the distribution parameters and fixed random seed, thus avoiding the need to store `FP16` auxiliary matrix.

### 4.3 Implementation of IntLoRA

Benefiting from the quantization-friendly weight distribution produced from the variance matching control, we provide two variants of our IntLoRA according to different quantizers on the adaptation term. The first variant employs the uniform quantizer on the adaptation term, thus enabling direct weight merge between the quantized adaptation term and pre-trained weights through integer multiplication. The second variant introduces log2 quantization to achieve downstream adaptation by bit-shifting the quantized pre-trained weights. More details are given below.

**Integer multiplication form.** We employ uniform affine quantization on the adaptation term, with the scaling factor and zero-point denoted as $\bar{s}$ and $\bar{z}$, and the quantized results as $\mathbf{U}_{\mathrm{round}}$, then our IntLoRA$_{\mathrm{MUL}}$ can be formalized as:

$$\mathbf{W}' = \bar{s} \cdot (\mathbf{U}_{\mathrm{round}} - \bar{z}) \odot (\mathbf{W}_{\mathrm{round}} - z). \tag{7}$$

**Bit-shifting form.** Denote the adaptation term in Eq. (3) as $\mathbf{V}$ for clarity, we first compute the bit shift value as follows:

$$\mathrm{shift} = \mathrm{clip}(\lfloor -\log_2 |\mathbf{V}| \rfloor, 0, 2^b - 1). \tag{8}$$

Then the weight adaptation with IntLoRA$_{\mathrm{SHIFT}}$ can be represented as:

$$\begin{aligned}
\mathbf{W}' &= \mathrm{sign}(\mathbf{V}) \odot 2^{-\mathrm{shift}} \odot (\mathbf{W}_{\mathrm{round}} - z) \\
&= \mathrm{sign}(\mathbf{V}) \odot [(\mathbf{W}_{\mathrm{round}} - z) \gg \mathrm{shift}] \\
&= \frac{1}{2^N} \odot \mathrm{sign}(\mathbf{V}) \odot [(\mathbf{W}_{\mathrm{round}} - z) \ll (N - \mathrm{shift})]
\end{aligned} \tag{9}$$

where $\mathrm{sign}(\mathbf{V}) \in \{-1, 1\}$ and $N = 2^b - 1$. Since the direct right-shifting on $\mathbf{W}_{\mathrm{round}} - z$ may lead to truncation error, we thus use $N - \mathrm{shift}$ with a scaling factor $\frac{1}{2^N}$ to equivalently convert to the left-shifting for error reduction.

## 5 Experiments

### 5.1 Experimental Setup

**Baselines.** Since there is little work studying PEFT of quantized diffusion models in integer arithmetic, we reproduce existing related approaches in LLMs and compare them with the proposed IntLoRA. Specifically, we include the following baselines: **i)** QLoRA (Dettmers et al., 2024), which fine-tunes the quantized model using `FP16` low-rank matrix. Since the adapted weights are still in `FP16`, we thus apply additional PTQ on the merged weights for a fair comparison. **ii)** QA-LoRA (Xu et al., 2023b), which uses group-shared low-rank parameters for seamless weight merge, and **iii)** IR-QLoRA (Qin et al., 2024), which improves the quantization with information retention as well as information elastic connection, and we also use additional PTQ on the merged `FP16` weights.

**Downstream tasks.** We evaluate different methods on multiple adaptation tasks, including subject-driven generation (Ruiz et al., 2023), controllable generation (Zhang et al., 2023), and style-customized image generation (Sohn et al., 2023). The subject-driven generation aims to generate images of the same subject given several images of a specific subject and a text prompt. We use

Table 1: Quantitative comparison on subject-driven generation tasks. We mark the proposed method with a gray background. The best results are **bolded**.

| Method | 8bit-bitwidth | | | | 4bit-bitwidth | | | |
|---|---|---|---|---|---|---|---|---|
| | DINO↑ | CLIP-I↑ | CLIP-T↑ | LPIPS↓ | DINO↑ | CLIP-I↑ | CLIP-T↑ | LPIPS↓ |
| LoRA-FP | 0.3625 | 0.7431 | 0.2499 | 0.7707 | 0.5625 | 0.7431 | 0.2499 | 0.7707 |
| QLoRA | 0.3635 | 0.6542 | 0.2550 | 0.7959 | 0.1564 | 0.5797 | 0.2260 | **0.7824** |
| QA-LoRA | 0.3635 | 0.6555 | 0.2537 | 0.7952 | 0.3412 | 0.6504 | 0.2514 | 0.7950 |
| IR-QLoRA | 0.3599 | 0.6547 | 0.2542 | 0.7960 | 0.2289 | 0.5907 | 0.2307 | 0.7829 |
| IntLoRA$_{MUL}$ | **0.3929** | **0.6721** | **0.2551** | 0.7928 | **0.3479** | **0.6546** | **0.2494** | 0.7997 |
| IntLoRA$_{SHIFT}$ | 0.3825 | 0.6641 | 0.2548 | 0.7930 | 0.3462 | 0.6478 | 0.2458 | 0.7970 |

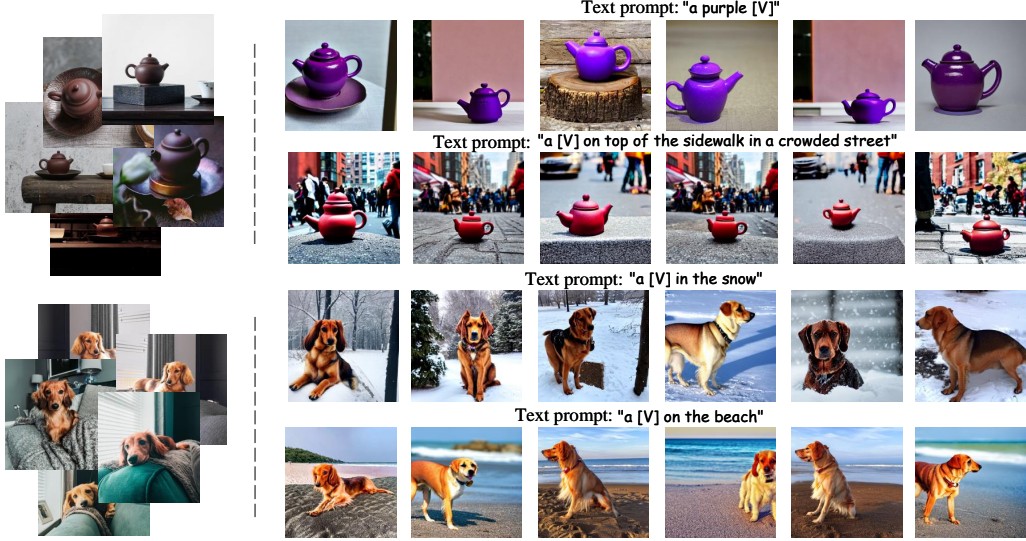

Figure 3: Quantitative comparison on subject-driven generation tasks. More results are provided in Fig. 14. Zoom in for better effects.

the dataset released by Dreambooth (Ruiz et al., 2023) for training and testing, which contains 25 subjects with each subject corresponding to 30 prompts. For controllable generation, we consider three sub-tasks, *i.e*, segmentation map to image on ADE20K dataset (Zhou et al., 2017), face landmark to image on CelebA-HQ dataset (Karras, 2017), and the canny edge to image on the COCO dataset (Lin et al., 2014). For the style-customized generation, we employ the StyleDrop (Sohn et al., 2023) dataset, which includes 18 style images, and we use 6 text prompts for each style to generate images with style similar to the style image and content aligned with the text prompt.

**Evaluation metrics.** In the subject-driven generation, we evaluate the quality of the generated images from three crucial aspects, *i.e.*, we use the DINO (Caron et al., 2021), CLIP-I (Radford et al., 2021) to access the subject fidelity, the CLIP-T (Radford et al., 2021) for textual prompt content fidelity, and LPIPS (Zhang et al., 2018) for sample diversity. For controllable generation task, we adopt both referenced and non-referenced image quality assessment metrics, namely FID (Heusel et al., 2017), CLIPIQA (Wang et al., 2023b), and NIQE (Mittal et al., 2012) to evaluate the performance of different methods. For style-customized generation, following StyleDrop (Sohn et al., 2023), we use CLIP-I to access the similarity between style images and generated images and use CLIP-T to evaluate the semantic fidelity between generated results and text prompts. However, as mentioned in StyleDrop (Sohn et al., 2023), these two scores are still not a perfect metric for performance evaluation, since even simply copying the style image as the generated results can also obtain a high CLIP-I score. To this end, we further adopt the ratio $\frac{\text{CLIP}-\text{I}}{\text{CLIP}-\text{T}}$ to comprehensively evaluate the balance between style and content fidelity.

**Implementation details.** We employ the StableDiffusionV1.5 (Rombach et al., 2022) as the pre-trained backbone for subject-driven generation and controllable generation. We further employ larger SDXL (Podell et al., 2023) as the pre-trained model in the style-customized generation. We use prevalent uniform affine quantization (Nagel et al., 2021) to quantize the pre-trained model

Table 2: Quantitative comparison on controllable image generation tasks.

| Method | #Bits | Segmentation-to-Image | | | Landmark-to-Face | | | Canny-to-Image | | |
|---|---|---|---|---|---|---|---|---|---|---|
| | | FID↓ | CLIPIQA↑ | NIQE↓ | FID↓ | CLIPIQA↑ | NIQE↓ | FID↓ | CLIPIQA↑ | NIQE↓ |
| LoRA | FP | 31.39 | 0.5710 | 4.45 | 37.50 | 0.5645 | 4.80 | 16.05 | 0.5582 | 5.37 |
| QLoRA | 8bit | 31.09 | 0.5669 | 4.44 | 35.73 | 0.5639 | 5.77 | 15.68 | 0.5901 | 5.16 |
| QA-LoRA | 8bit | 31.32 | 0.5755 | **4.17** | 38.88 | 0.5667 | 5.56 | 15.34 | 0.6001 | 5.32 |
| IR-QLoRA | 8bit | 31.81 | **0.5761** | 4.30 | 36.30 | 0.5683 | 5.74 | 15.70 | 0.6005 | **4.72** |
| IntLoRA$_{MUL}$ | 8bit | **31.08** | 0.5693 | 4.43 | 37.52 | **0.5686** | 5.67 | **15.26** | **0.6013** | 4.96 |
| IntLoRA$_{SHIFT}$ | 8bit | 31.38 | 0.5703 | 4.22 | **34.46** | 0.5495 | **5.19** | 15.76 | 0.5913 | 4.83 |
| QLoRA | 4bit | 71.75 | 0.5584 | 4.61 | 117.37 | 0.5578 | 4.84 | 62.49 | 0.5728 | 4.70 |
| QA-LoRA | 4bit | 31.51 | 0.5646 | 4.31 | 43.09 | 0.5748 | 5.00 | 16.73 | 0.5745 | **4.52** |
| IR-QLoRA | 4bit | 35.83 | 0.5624 | 4.57 | 39.63 | 0.5632 | 5.23 | 18.30 | 0.5733 | 4.58 |
| IntLoRA$_{MUL}$ | 4bit | **31.27** | 0.5650 | **4.16** | **33.62** | 0.5667 | **4.55** | **16.32** | 0.5746 | 4.55 |
| IntLoRA$_{SHIFT}$ | 4bit | 32.85 | **0.5693** | 4.57 | 43.64 | **0.5819** | 5.81 | 17.65 | **0.5790** | 4.60 |

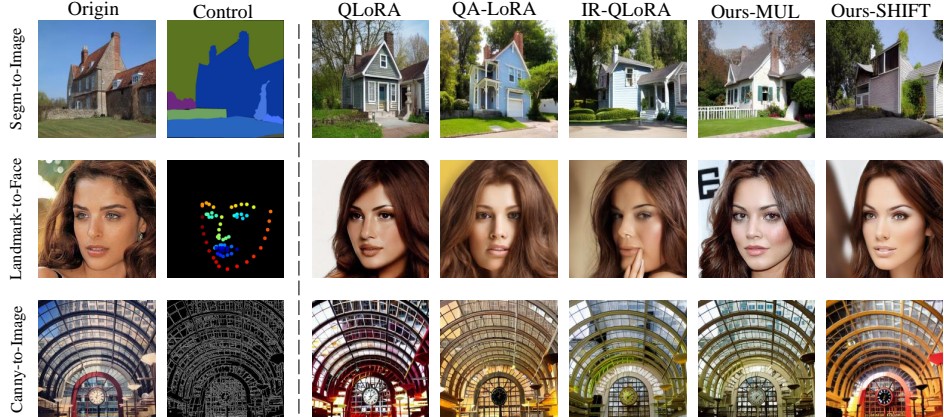

Figure 4: Quantitative comparison on controllable generation tasks. More results are provided in Appendix O. Zoom in for better effects.

weights. We append the trainable low-rank parameters on the Query, Key, Value, and Out projection in the attention layers (Vaswani et al., 2017) and keep all other layers frozen and quantized. The rank of LoRA is set to 4 for subject-driven generation and controllable generation, and 64 for style-customized generation. For compared baselines, we control the rank to ensure a fair comparison. As for the variance matching control, we employ the ratio of the maxima of the sampled distributions as a fast estimator of the variance ratio. As for the training of the quantized adaptation term, we use the Straight Through Estimator (STE) on the quantized adaptation term to allow backpropagation. Given that backpropagation with the integer type can severely impair performance, we mainly consider simulated quantization during fine-tuning, while keeping the standard integer format for storage and inference purposes. Due to the page limit, we provide more details in Appendix I.

## 5.2 MAIN RESULTS AND EFFICIENCY

**Subject driven generation.** We first use our IntLoRA to fine-tune the pre-trained models on the subject-driven generation task. This task requires a few optimization steps for fine-tuning, and the downstream weights do not deviate too much from the pre-trained capabilities. Therefore, it can reflect the short-term tuning ability of different methods. Tab. 1 shows the results under 8-bit and 4-bit weight quantization settings. It can be seen that the proposed methods consistently outperform other competitors under different bit-width stetting. For instance, the IntLoRA$_{MUL}$ suppresses the IR-QLoRA by 0.1190 DINO score on the 4-bit setup. Notably, the QLoRA and IR-QLoRA baseline uses additional PTQ on the fine-tuned model, leading to a significant performance drop under the 4-bit width. In contrast, even the log2-quantization of our IntLoRA$_{SHIFT}$ works well under 4-bit thanks to the proposed VMC. We also give qualitative visualization in Fig. 3, where one can see that our IntLoRA can facilitate subject-faithful and photo-realistic image generation.

**Controllable image generation.** We further validate the effectiveness of IntLoRA on the controllable image generation. Since this task requires more tuning steps for adaptation in input spaces different from pre-training, it can represent the long-term adaptation ability of different methods.

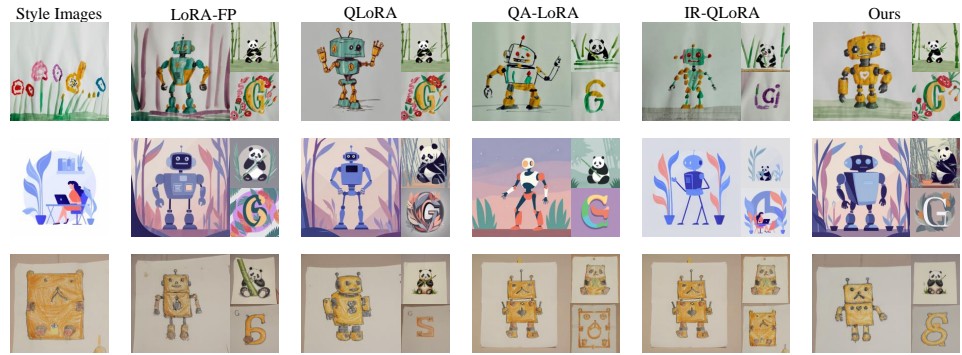

Figure 5: Quantitative comparison on style-customized generation. The text prompt is "A friendly robot in [V] style", "A panda eating bamboo in [V] style", and "The letter 'G' in [V] style", respectively. 'Ours' denotes the IntLoRA$_{\mathrm{SHIFT}}$. More results are provided in Fig. 18.

Table 3: Quantitative comparison on style-customized image generation tasks. The ratio is $\frac{\mathrm{CLIP-I}}{\mathrm{CLIP-T}}$ to evaluate style-content balance.

| Metrics | 8bit-bitwidth | | | 4bit-bitwidth | | |
|---|---|---|---|---|---|---|
| | CLIP-I | CLIP-T | ratio | CLIP-I | CLIP-T | ratio |
| LoRA-FP | 0.6010 | 0.2582 | 2.33 | 0.6010 | 0.2582 | 2.33 |
| QLoRA | 0.6016 | 0.2583 | 2.33 | 0.5466 | 0.2571 | 2.13 |
| QA-LoRA | 0.6593 | 0.2381 | 2.77 | 0.6086 | 0.2373 | 2.56 |
| IR-QLoRA | 0.7025 | 0.2198 | 3.20 | 0.5957 | 0.2365 | 2.52 |
| Ours-MUL | 0.5832 | 0.2596 | 2.24 | 0.5711 | 0.2551 | 2.23 |
| Ours-SHIFT | 0.6043 | 0.2569 | 2.35 | 0.5708 | 0.2566 | 2.22 |

Table 4: Efficiency comparison under SDXL backbone 4-bit quantization setting. #param denotes the number of trainable parameters. $N$ is the downstream task number.

| Methods | #param(M) | storage(GB) | PTQ | weight merge |
|---|---|---|---|---|
| LoRA-FP | 92M | 9.8+0.34N | ✔ | FP addition |
| QLoRA | 92M | 1.2+0.34N | ✔ | FP addition |
| QA-LoRA | 85M | 1.2+0.34N | ✘ | FP addition |
| IR-QLoRA | 85M | 1.2+0.34N | ✔ | FP addition |
| IntLoRA$_{\mathrm{MUL}}$ | 92M | 1.2+0.04N | ✘ | int-mul |
| IntLoRA$_{\mathrm{SHIFT}}$ | 92M | 1.2+0.04N | ✘ | bit-shift |

The results, shown in Tab. 2, indicate that our method achieves comparable results against existing strong baselines, *e.g.*, our IntLoRA$_{\mathrm{MUL}}$ outperforms the IR-QLoRA by 4.56 FID score. Fig. 4 provides the qualitative comparison on three controllable image generation tasks, and it can be seen that the images generated by the IntLoRA-tuned model are well matched with the control signals.

**Style customized generation.** Tab. 3 shows the results of our IntLoRA compared with other baselines on the style customized generation task. Taking the ratio $\frac{\mathrm{CLIP-I}}{\mathrm{CLIP-T}}$ of the LoRA-FP model as the metric reference, our method achieves a similar balance ratio of style and content as the LoRA-FP. In contrast, QA-LoRA and IR-QLoRA overfit to specific style images, reflecting excessively high CLIP-I and low CLIP-T. We also provide visualization results in Fig. 5. It can be seen that our IntLoRA achieves a favorable balance between style images and text prompts, whereas some existing approaches fail, *e.g.*, the third row and fifth column with prompt "The letter 'G' in [V] style", where the generated images almost copy the original style image.

**The efficiency of IntLoRA.** Although the ultimate goal of all compared methods is the quantized diffusion models for fast inference, our IntLoRA can achieve this target with less cost. Specifically, as shown in Tab. 4, a clear advantage of our method lies in its seamless switch between tuning and inference, without additional PTQ. In addition, our IntLoRA is the only one that achieves `INT4` type adaptation weights, significantly reducing the storage costs when there are many downstream tasks. Furthermore, our methods allow for weight merge using the hardware-friendly operator, naturally enabling the merged weights still in `INT4` to facilitate the employment of edge devices.

## 5.3 ABLATION STUDIES

In this section, we study the effectiveness of different components in the proposed IntLoRA. The experiments are conducted with IntLoRA$_{\mathrm{SHIFT}}$ on the subject-driven generation with 15 subject-prompt pairs under 4-bit weight quantization.

**Contradictory aspect of auxiliary variances.** In Sec. 4.3, we have theoretically pointed out that there is a choice dilemma for $\sigma_{\mathbf{R}}$. Here we elaborate on its effect through distribution visualization. Specifically, we remove the VMC and use a scaling scalar to generate a too-large or too-small auxiliary variance, followed by the log2 quantization on the adaptation term. The results are shown in Fig. 6. On the one hand, setting $\sigma_{\mathbf{R}}$ too large can lead to a low correlation $\rho(\mathbf{W}, \mathbf{W} - \mathbf{R})$, which makes it hard to reconstruct $\mathbf{W}$ from $\mathbf{W} - \mathbf{R}$ using estimator $\mathbf{W} \approx \mathcal{Q}(\mathbf{W} - \mathbf{R}) + \mathbf{R}$. On the

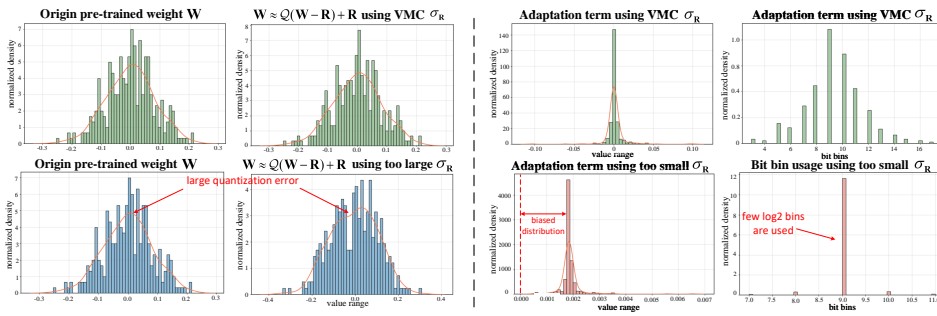

Figure 6: The distribution visualization using Kernel Density Estimate (KDE) on different weight tensors. **Left**: the KDE plot of pre-trained weights and estimated weights under different $\sigma_{\mathbf{R}}$. **Right**: the KDE plot of the adaptation term and the log2 bins usage with different $\sigma_{\mathbf{R}}$.

other hand, a too small $\sigma_{\mathbf{R}}$ prevents the expectation of adaptation term converging to zero, causing few log bins to be used. In experiments, we find that the training of both settings fails to converge. By contrast, the proposed VMC can precisely control $\sigma_{\mathbf{R}}$ to allow most values of the adaptation term to be zero-neighbored, facilitating more challenging log2 quantization. Moreover, it should be noted that the too small $\sigma_{\mathbf{R}}$ can also be regarded as an approximation of direct quantization on the zero-initialized $\mathbf{AB}$, and thus the experimental results also justify the AQS for zero-initialized $\mathbf{AB}$.

**Distribution selection for auxiliary matrix.** In the proposed IntLoRA, the auxiliary matrix $\mathbf{R}$ plays a crucial role in both AQS and VMC. Therefore, the distribution shape of $\mathbf{R}$ can greatly influence the performance. To this end, we ablate different distributions as the instantiation of $\mathbf{R}$ and give the corresponding performances in Tab. 5. It can be seen that the Laplace distribution performs better than the other options. This is because a light-tailed distribution, such as Laplace, clusters most samples around zero, which is consistent with the re-

Table 5: Ablation on different distribution choices of the auxiliary matrix.

| settings | DINO↑ | CLIP-I↑ | CLIP-T↑ | LPIPS↑ |
|---|---|---|---|---|
| Guassian | 0.4335 | 0.6956 | 0.2492 | 0.8179 |
| Cauchy | 0.1367 | 0.5617 | 0.1870 | 0.8067 |
| StudentT | 0.2935 | 0.6420 | 0.2587 | 0.8048 |
| Laplace | 0.3992 | 0.6680 | 0.2548 | 0.8311 |

quirement for log2-quantization. And this accurate log2-quantization further facilitates favorable performance. Therefore, the light-tailed distribution performs experimentally better than the heavy-tailed counterparts. For better effects, we provide the shape of different distributions in Fig. 12.

**Ablation on the smoothing factor.** In order to produce quantization-friendly adaptation term, we propose the VMC which rescales $\sigma_{\mathbf{R}}$ by $r^{\alpha}$, where $r$ is the variance ratio $\frac{\sigma_{\mathbf{W}}}{\sigma_{\mathbf{R}}}$ and $\alpha$ is a hyperparameter to search for the optimal control. Given $r < 1$ (the distribution of $r$ is given in Fig. 10), making $\alpha < 1$ further scales up $\sigma_{\mathbf{R}}$, which leads to lowered correlation between $\mathbf{W}$ and $\mathbf{W} - \mathbf{R}$, making it difficult to recover $\mathbf{W}$ from $\mathbf{W} - \mathbf{R}$. Moreover, setting $\alpha > 1$ can decrease $\sigma_{\mathbf{R}}$, making the distribution being biased according to Eq. (5). Experiments in Fig. 7 show that setting $\sigma_{\mathbf{R}}$ slightly smaller than $\sigma_{\mathbf{W}}$ can obtain favorable performance, in-

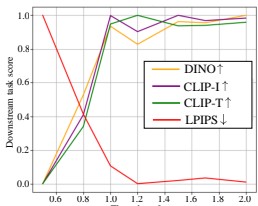

Figure 7: The normalized performance under different $\alpha$.

dicating that the information loss of $\mathbf{W}$ has a greater impact than a biased adaptation term. In the implementation, we chose a moderate smoothing factor $\alpha = 1.5$ for the trade-off.

## 6 CONCLUSION

We propose IntLoRA, a novel joint adaptation and quantization framework which can address the arithmetic inconsistency by employing integer low-rank parameters, to push the efficiency limits of diffusion model fine-tuning. We propose the quantization-adaptation separation to allow the coexistence of zero-initialized gradient and quantization-friendly distribution. We further introduce the multiplicative low-rank adaptation to achieve a decoupled quantizer of pre-trained and adaptation weights for PTQ-free weight merge, accompanied by the variance matching control to adjust the channel-aware variance for accurate adaptation control. Benefiting from the elegant design, we provide two variants of IntLoRA, which either use int-multiplication or bit-shifting to adapt the quantized pre-trained models. Through transferring the adaptation weights to the integer arithmetic, our IntLoRA demonstrates its effectiveness across different pre-trained models and various downstream tasks, while exhibiting impressive efficiency across model tuning, storage, and inference.

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

# APPENDIX

## A  EVALUATION ON NATURAL LANGUAGE

In the main paper, we mainly validate the effectiveness of our IntLoRA on the image generation tasks. In this section, we turn our focus to the natural language to give a broader view of performance comparison. Specifically, we apply different methods to fine-tune the Llama3-8B model (Dubey et al., 2024) and use the MetaMathQA (Yu et al., 2023) dataset for training and GSM8K (Cobbe et al., 2021) dataset for testing. In all experiments, we fine-tune the model using the AdamW optimizer for 2 epochs with the cosine learning scheduler, and the warm-up ratio of the learning scheduler is set as 0.005. We follow the evaluation tools in MetaMathQA, which use the Alpaca prompt and evaluate the model in zero-shot. Due to the GPU memory limitation, we only append the LoRA layer to the attention query and value projection matrices. The comparison results are shown in Appendix O.1. It can be seen that our IntLoRA matins stable performance when transferring to natural language tasks. We also provide some testing cases in Appendix O for better presentation.

## B  INTLORA WITH ACTIVATION QUANTIZATION

We further supplement the activation quantization of our IntLoRA. In detail, we employ the simple per-tensor Uniform Affine Quantization on the activation, without using other complicated tricks, and find it could work well on up to W4A8 quantization setting. The results of Dreambooth fine-tuning are shown in Tab. 7. It can be seen that our IntLoRA maintains superior performance under activation quantization settings. Importantly, the addition of activation quantization allows our IntLoRA to achieve practical inference speed-up. As for the W4A4 setup, we observe both our IntLoRA and all other baselines fail, which suggests additional techniques are needed for lower activation quantization bits. Considering that we mainly aim to address the challenge of arithmetic inconsistency during weight merge, we leave further 4-bit activation quantization for future work.

## C  INTLORA FOR DIFFUSION MODEL QUANTIZATION

Since our IntLoRA can potentially be used in diffusion model quantization as well, we also apply our IntLoRA to the difusion model quantization experimental settings. The experimental results are shown in Tab. 8. It can be seen that our IntLoRA$_{\text{MUL}}$ achieves even better performance than the EfficientDM (He et al., 2023) on the W4A4 quantization experimental setting. It should also be noted that our IntLoRA only need to load the quantized model during the calibration instead of the floating-point weights in EfficientDM, thus reducing the memory cost. The above results show that our IntLoRA can be potentially applied to the network quantization, demonstrating the generality of the proposed method on the diffusion quantization tasks.

## D  COMPARISON WITH EFFICIENTDM

In this section, we compare in detail with the EfficientDM (He et al., 2023), which employs LoRA for the diffusion network quantization task, to provide a clearer benchmark of IntLoRA's performance against existing methodologies with similar strategies.

Firstly, EfficientDM is more suitable for quantization, since directly quantizing the merged weights is sufficient given quantization aims to obtain *one* quantized model. For downstream task adaptation, where *multiple* quantized downstream models are required, the proposed IntLoRA is more suitable for downstream adaptation, which allows separately quantized weights and adapters as well as seamless weight merge. Secondly, EfficientDM focuses on the quantized merged weights, and an obvious drawback is that it still needs to load the large `FP32` weights during fine-tuning, which is not suitable for user customization on consumer-level GPUs. For example, just loading the `FP32`

Table 6: Quantitative comparison on the natural language tasks of mathematical answering.

| Methods | LoRA-FP | QLoRA | QA-LoRA | IR-QLoRA | IntLoRA$_{MUL}$ | IntLoRA$_{SHIFT}$ |
|---|---|---|---|---|---|---|
| #praram | 1.17M | 1.17M | 1.18M | 1.18M | 1.17M | 1.17M |
| accuracy | 62.24% | 64.06% | 63.53% | 63.00% | 64.23% | 64.10% |

Table 7: Quantitative comparison of the weight-activation quantization setting on the Dreambooth task. We use the average results of 15 text-subject pairs for evaluation.

| methods | nbits(W/A) | DINO↑ | CLIP-I↑ | CLIP-T↑ | LPIPS↓ |
|---|---|---|---|---|---|
| LoRA-FP | W32A32 | 0.4828 | 0.6968 | 0.2954 | 0.8076 |
| QA-LoRA | W8A8 | 0.4156 | 0.6664 | 0.2834 | 0.8086 |
| IR-QLoRA | W8A8 | 0.4070 | 0.6630 | 0.2841 | 0.8110 |
| IntLoRA$_{MUL}$ | W8A8 | 0.4498 | 0.6882 | 0.2858 | 0.8062 |
| IntLoRA$_{SHIFT}$ | W8A8 | 0.4353 | 0.6842 | 0.2841 | 0.8257 |
| QA-LoRA | W4A8 | 0.4127 | 0.6897 | 0.2700 | 0.8281 |
| IR-QLoRA | W4A8 | 0.3722 | 0.6719 | 0.2707 | 0.8086 |
| IntLoRA$_{MUL}$ | W4A8 | **0.4242** | **0.6913** | **0.2710** | **0.8181** |
| IntLoRA$_{SHIFT}$ | W4A8 | 0.4039 | 0.6716 | 0.2709 | 0.8147 |

FLUX.1-dev can consumes over 23GB GPU memory. Thirdly, each downstream task needs to store the corresponding large quantized model when using EfficientDM for adaptation, which leads to noticeable challenge for customized model sharing among users like Civitai. As a comparison, our IntLoRA only need to store 1 quantized pre-training weight and per-task quantized adapter weights to reduce storage cost.

# E  THE SIGNIFICANCE OF ACCELERATED TRAINING

In the downstream adaptation, since the the pre-trained weights is fixed and only small adapters are updated during fine-tuning, it naturally rises questions of the importance of the speed-up for the training stage in our IntLoRA. Here, we would like to point out that the original LoRA (Hu et al., 2021) only reduces the GPU memory of *gradient* and *optimizer states* during training, and cannot benefit the model weights. Considering that the diffusion models are getting bigger, *e.g.*, just importing FLUX.1-dev to cuda can take up 23.8GB, it makes sense to reduce the size of the weights during training. Our IntloRA only needs the quantized pre-training weights during training, further facilitates user-customized diffusion fine-tuning on consumer-level GPUs. In detail, taking SDXL (Podell et al., 2023) as an example, full fine-tuning SDXL consumes 59GB GPU memory, of which 13.5GB comes from the weights. Using `FP32` LoRA tuning consumes 28.38GB, indicating 30.62GB reduced gradients and optimizer states. However, this LoRA tuning still maintains the weights in `FP32`. Theoretically, our IntLoRA can further reduce this 13.5GB `FP32` weight to 3.375GB with the `INT8` quantization. Finally, efficient training is only one of the benefits of our IntLoRA, whose another advantage is to obtain downstream quantized models without additional PTQ for efficient inference.

# F  DISCUSSION ON TRAINING EFFICIENCY

We compare the training speed of our IntLoRA against other baselines in Tab. 9. It can be seen that our IntloRA exhibits slightly larger training costs than LoRA-FP. This makes sense due to the fact that QAT typically consumes more cost than the common training counterpart. However, the drawback of LoRA-FP is that it is difficult to speed up in the inference phase. Moreover, comparing

Table 8: Quantitative comparison with EfficientDM (He et al., 2023) on diffusion model quantization tasks. We use the ImageNet $256 \times 256$ image generation tasks, and train the ddim_step=20 LDM-4 model with 500 training epochs.

| methods | nbits(W/A) | IS↑ | FID↓ | sFID | precision↓ | recall↑ |
|---|---|---|---|---|---|---|
| EfficientDM | W4A4 | 178.20 | 13.42 | 26.67 | 0.70 | 0.58 |
| IntLoRA-SHIFT | W4A4 | 116.50 | 20.20 | 26.79 | 0.63 | 0.58 |
| IntLoRA-MUL | W4A4 | **199.20** | **10.43** | **24.02** | **0.79** | **0.55** |

Table 9: Comparison of training efficiency with other methods. We fine-tuning the Stable Diffusion 1.5 model on the Dreambooth task. The training speed is tested on one NVIDIA 3090 GPU.

| setup | traing-speed | GPU-usage | Inference acceleration | Additional PTQ |
|---|---|---|---|---|
| Full-finetune | 0.74s/img | 19.4G | No | Yes |
| LoRA-FP | 0.68s/img | 6.7G | No | Yes |
| QLoRA(W8-only) | 0.71s/img | 6.9G | No | Yes |
| QLoRA(W8A8) | 0.85s/img | 7.2G | Yes | Yes |
| IntLoRA(W8-only) | 0.72s/img | 7.1G | No | No |
| IntLoRA(W8A8) | 0.87s/img | 7.4G | Yes | No |

with QLoRA, our IntloRA achieves similar training complexity, but our approach does not require additional PTQ, facilitating user-customized local deployment.

## G  FURTHER EXPLANATION OF THE AUXILIARY MATRIX.

In the proposed method, the auxiliary matrix $\mathbf{R}$ is crucial to ensure quantization-friendly distribution and meaningful optimization trajectories. Here, we summarize two aspects of the role of $\mathbf{R}$ in the proposed method. **First**, $\mathbf{R}$ can be viewed as an initialization parameter for trainable parameters $\mathbf{AB}$. Specifically, we use $\mathbf{R}$ without gradient in the proposed AQS to guarantee easily quantized adaptation distributions while guaranteeing zero-initialized gradient trajectories. **Second**, the presence of $\mathbf{R}$ in the adaptation term also implies that it can be used to control the distribution of the adaptation term, thus allowing for more challenging quantizers, such as log2 quantization. For this, we propose the VMC, which makes the expectation of the adaptation term converge to zero, enabling as many log-scale buckets to be used as possible. Despite this indispensable role of $\mathbf{R}$, finding the optimal shape and variance of $\mathbf{R}$ remains challenging. In this work, we attempt to solve this from an engineering perspective by ablating different distributions and the variance scales. Experimental results demonstrate the effectiveness of this solution.

## H  DIFFERENCES FROM QA-LORA

In this section, we give a detailed discussion about the differences between the proposed IntLoRA and QALora (Xu et al., 2023b). **First**, one of the limitations of QA-LoRA is that it requires the pre-trained weights and the adaptation weights to share the same quantizer, which leads to a discrete parameter search space, limiting the adaptation capability for downstream tasks. In contrast, our IntLoRA ensures a more powerful adaptation through the proposed MLA which reformulates the form from additive to multiplicative, thus allowing independent quantizers. **Second**, the adaptation weights of QALoRA are still in `FP16`, leading to storage cost as well as `FP16` addition for weight merge during deployment, whereas our IntLoRA only needs to store `INT4` weights while fusing weights through hardware-efficient integer multiplication or bit shifting. **Third**, QALoRA requires additional group quantization in the input channel dimensions, leading to additional `GEMM` core design, while our approach uses a common quantization strategy, thus facilitating practical deployment.

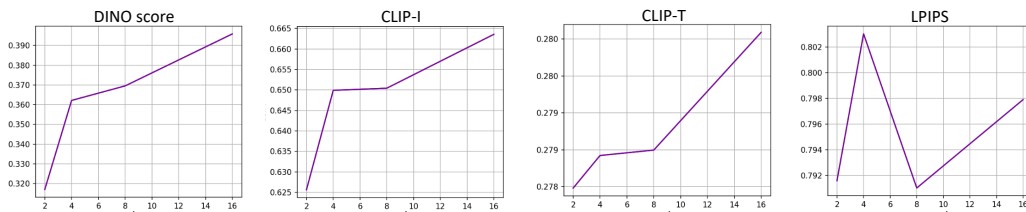

Figure 8: Ablation experiments of different LoRA ranks.

# I  MORE IMPLEMENTATION DETAILS

For the subject-driven generation, we use the AdamW optimizer with a weight decay of 1e-2 and fine-tune the query, key, value, and output projection layer. The learning rate is set to 6e-5. The batch size is set to 1, and the number of training steps is 400. The rank of the LoRA is set to 4. We adopt the prior preservation strategy as Dreambooth (Ruiz et al., 2023) to generate 200 class images. For the controllable generation, we fine-tune the model for 11 epochs for Canny-to-Image tasks and 20 epochs for Landmark-to-Face and Segmentation-to-Image tasks. The learning rate is set to 1e-5 using the AdamW optimizer. The LoRA rank is set to 4. The batch size is set to 8 and the image resolution is $512 \times 512$ for all three tasks. For the style-customized generation, we fine-tune the pre-trained model using the AdamW optimizer with a learning rate of 5e-5. Since it involves a larger SDXL, we chose a relatively large LoRA rank of 64 for all compared methods, since there is only one style image as well as the larger pre-trained parameters. We fine-tune for 500 steps with batch size 1. Similar to StyleDrop (Sohn et al., 2023), we only use one image as the style image and find it works well. The style images and text prompts for evaluation are given in Appendix O. The variance ratio in the variance matching control is surrogated as the value range ratio, *i.e.*, $r = \frac{\max\{|\max(\mathbf{W})|,|\min(\mathbf{W})|\}}{\min\{|\max(\mathbf{R})|,|\min(\mathbf{R})|\}}$.

# J  ADDITIONAL ABLATION EXPERIMENTS

**Ablation on the LoRA rank.** The low-rank $d$ in LoRA is a trade-off between performance and efficiency. A larger rank improves the adaptation ability by training more parameters but comes with larger training and storage costs, and vice versa. Here, we give the impact of different rank setups on performance in Fig. 8. One can see that the performance generally improves as we increase the rank, but the rate of growth varies. For instance, the increase speed from rank=4 to rank=8 increases inferior to the one of from rank=2 to rank=4. Moreover, increasing the rank to 16 can generally obtain better results than its lower counterpart. In practice, considering the trade-off between performance and efficiency, we select a moderate rate rank=4.

**The effects of variance matching control for IntLoRA$_{\mathrm{MUL}}$.** In this work, we propose the variance matching control to adjust the variance of $\mathbf{R}$, so that allows the log2-quantization of the adaptation term to obtain the IntLoRA$_{\mathrm{SHIFT}}$. In other words, the VMC is primarily introduced for IntLoRA$_{\mathrm{SHIFT}}$. Despite we also apply the VMC to IntLoRA$_{\mathrm{MUL}}$, given the IntLoRA$_{\mathrm{MUL}}$ does not require such strict constraints on the distribution shape of the adaptation term, it is interesting to investigate the influence of variance matching control on the performance of IntLoRA$_{\mathrm{MUL}}$. To this end, we adjust the smoothing factor $\alpha$ to adjust the strength of the VMC, *e.g.*, setting $\alpha$ to zero can lead to the removal of the VMC. The results of the

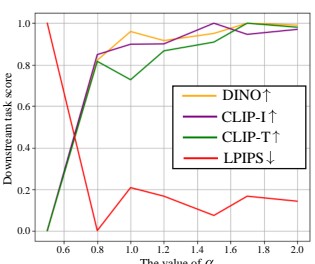

Figure 9: The effects of VMC for IntLoRA$_{\mathrm{MUL}}$.

IntLoRA$_{\mathrm{MUL}}$ under different VMC scales are shown in Fig. 9. As one can see, despite the VMC being initially proposed for the log2-quantization, the well-structured distribution also facilitates uniform quantization. For example, when we set the $\alpha$ approaching to zero, i.e. the VMC is close to being removed, the performance of IntLoRA$_{\mathrm{MUL}}$ appears similar pattern as the IntLoRA$_{\mathrm{SHIFT}}$, which suffers a significant performance drop. Moreover, the performance gains gradually converge

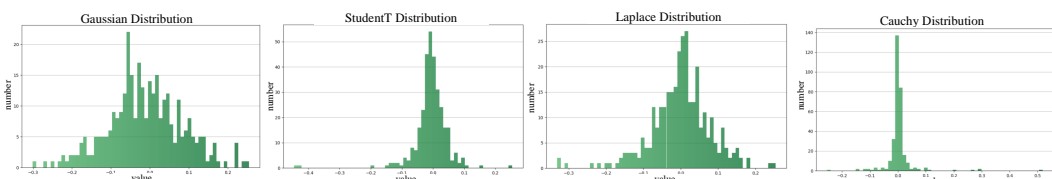

Figure 11: The value ranges of different channels in the pre-trained weights.

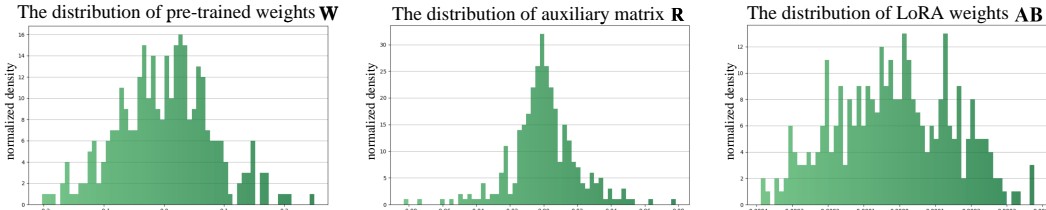

Figure 12: The shape of different distributions for initialing the auxiliary matrix.

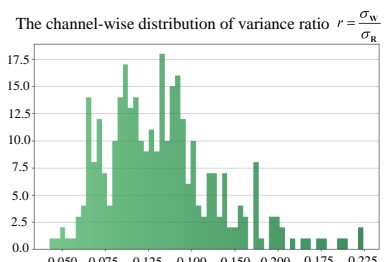

Figure 13: The distribution visualization of the original weights $\mathbf{W}$, the auxiliary matrix $\mathbf{R}$, and the learned low-rank weights $\mathbf{AB}$.

when the $\alpha > 1.5$. In short, the VMC can not only allow the log2-quantization to work but also improve the performance of the uniform quantization.

**Distribution shape for auxiliary matrix.** In Sec. 5.3, we provided different symmetric distributions including Gaussian, StudentT, Laplace, and Cauchy. Fig. 12 gives the results of sampling from different distributions. The Laplace distribution possesses light tails, and the shape of the distribution is convex, *i.e.*, $f''(x) > 0, x \neq 0$. This unique property makes it easy to control the value of the adaptation term to produce distributions that are friendly to log2 quantization, *i.e.*, most samples are clustered around the zero to use as many bins as possible. This analysis is also verified by the experiments in Tab. 5, which shows that the Laplace distribution achieves the best performance.

**Reasons behind channel-wise control in VMC.** In the main paper, we study how to control the auxiliary matrix to produce a compatible $\sigma_{\mathbf{R}}$ by searching for the $r^\alpha$. As for the variance ratio $r$, it is worth noting that we adopt a per-channel scaling factor $r \in \mathbf{R}^{C_{out}}$ instead of the per-tensor one. Here we elaborate on the reasons for this. As shown in Fig. 11, we give a boxplot of the pre-trained weight $\mathbf{W}$ and it can be seen that the values for each channel exhibit different variances. Therefore, considering that the auxiliary matrix has a large impact on performance, we opt for this fine-grained channel-wise control. In addition, since $r$ is task-agnostic, it can be computed in advance and stored once. Finally, we give the distribution of $r$, and one can see that $r$ is generally less than 1, with a single peak and a left-skewed distribution shape.

Figure 10: The value distribution of the channel-wise variance ratio $r$

## K    IMPACTS FROM THE AUXILIARY MATRIX

In Eq. (2) of the proposed AQS, we introduce an additional auxiliary matrix $\mathbf{R}$ to the original pre-trained weight $\mathbf{W}_0$ to achieve adaptation-quantization separation. However, this extra $\mathbf{R}$ potentially introduces outliers and thus cause quantization error for $\mathbf{W}$. Here, we point out that since the proposed VMC can control the range of $\mathbf{R}$ through the variance scaling factor $r = \sigma_{\mathbf{W}}/\sigma_{\mathbf{R}}$, the introduction of $\mathbf{R}$ in the AQS is ensured not result in additional outliers. For validation, we also give the distribution visualization of the original $\mathbf{W}$ and the VMC re-scaled $\mathbf{R}$ in Fig. 13. It can be seen that the range of $\mathbf{R}$ is effectively controlled within the range of $\mathbf{W}$, thus effectively avoiding the detrimental effect of additional outliers.

## L    JUSTIFICATION FOR THE VALUE ORDERS

A key assumption in the derivation for VMC is that the learned values of low-rank parameters $\mathbf{AB}$ is orders of magnitude smaller than the auxiliaty matrix $\mathbf{R}$. Based on this assumption we ignore $\mathbf{AB}$ as an approximation. Here, we give the specific evidence for this approximation. Specifically, we visualize the weights of the trained $\mathbf{AB}$ and the distribution of $\mathbf{R}$, as shown in Fig. 13. It can be seen the range of $\mathbf{AB}$ is constrained to $[-0.0004, 0.0004]$, while the range of $\mathbf{R}$ is $[-0.08, 008]$. Therefore, the experimental visualization above confirms the soundness of our approximation. Since the $\mathbf{AB}$ in LoRA is zero initialized, it tends to be distributed around zero with learned small values aiming to not disturb the pre-training weights too much.

## M    ADVANTAGE DISCUSSION OF INTLORA

Here, we would like to discuss the advantages of our InLoRA, to highlight the contribution of this work. **First**, since the downstream fine-tuning is performed on the quantized pre-trained weights, and thus the adaptation pipeline with our IntLoRA only needs to load the quantized weights, reducing memory footprint from `FP16` weights. **Second**, transferring the adaptation parameters to also the `INT4` arithmetic allows all weights of the model can be stored in `INT4` format, reducing the disk space for storage. **Thirdly**, thanks to the designed multiplicative adaptation form, our IntLoRA enables seamless weight merge of quantized pre-trained weights and adaptation weights at test time without additional post-training quantization, streamlining the adaptation process while alleviating performance drop from quantization.

## N    LIMITATION AND FUTURE WORKS

Although the proposed IntLoRA can effectively improve the efficiency of diffusion model fine-tuning by allowing the adaptation parameters also on the integer arithmetic, the proposed framework can be further improved in the following aspects. First, although the trainable low-rank weights are quantized with STE, these quantized weights are still in `FP16` type during tuning to enable accurate gradient updates. Therefore, it is promising to specifically design integer-type propagation. Despite this seems challenging, it can further reduce the training cost and accelerate the adaptation process. Second, although we introduce a feasible way that uses hyperparameter search of the smoothing factor $\alpha$ to find a compatible $\sigma_{\mathbf{R}}$ as well as the appropriate distribution shape of $\mathbf{R}$, it can be more elegant if we can use advanced mathematical analysis techniques, such as functional analysis, to find the statistical properties a suitable $\mathbf{R}$ should satisfy. Third, this work mainly focuses on the efficient acceleration of LoRA due to its prevalence among the PEFT techniques. Applications to other PEFT methods for hardware-efficient adaptation could be interesting future work.

## O    ADDITIONAL VISUALIZATION RESULTS

- Fig. 14 gives more samples on the subject-driven generation tasks.
- In Fig. 15, we give more samples of the segmentation-to-image tasks.
- In Fig. 16, we give more samples of the face landmark-to-face image tasks.

- In Fig. 17, we give more samples of the canny edge-to-image tasks.

- Fig. 18 provides more samples of the results of the style-accustomed generation.

- In Fig. 19, we give the style images and the text prompts used for evaluation on the style customized generation tasks.

- In Appendix O.1, we give some case studies of the mathematical question-answering task using the fine-tuned Llama3-8B model.

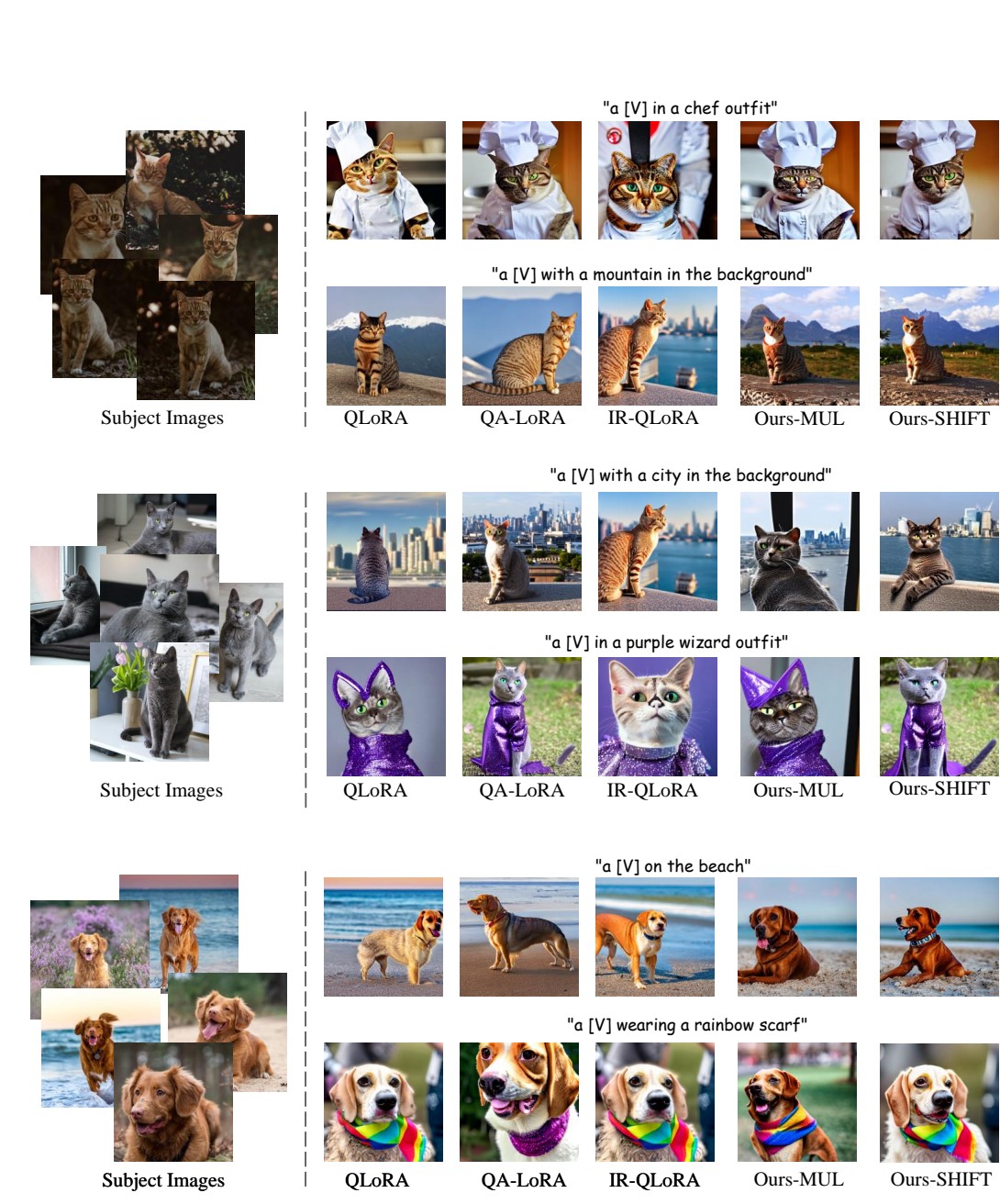

Figure 14: More qualitative comparison results on subject-driven generation.

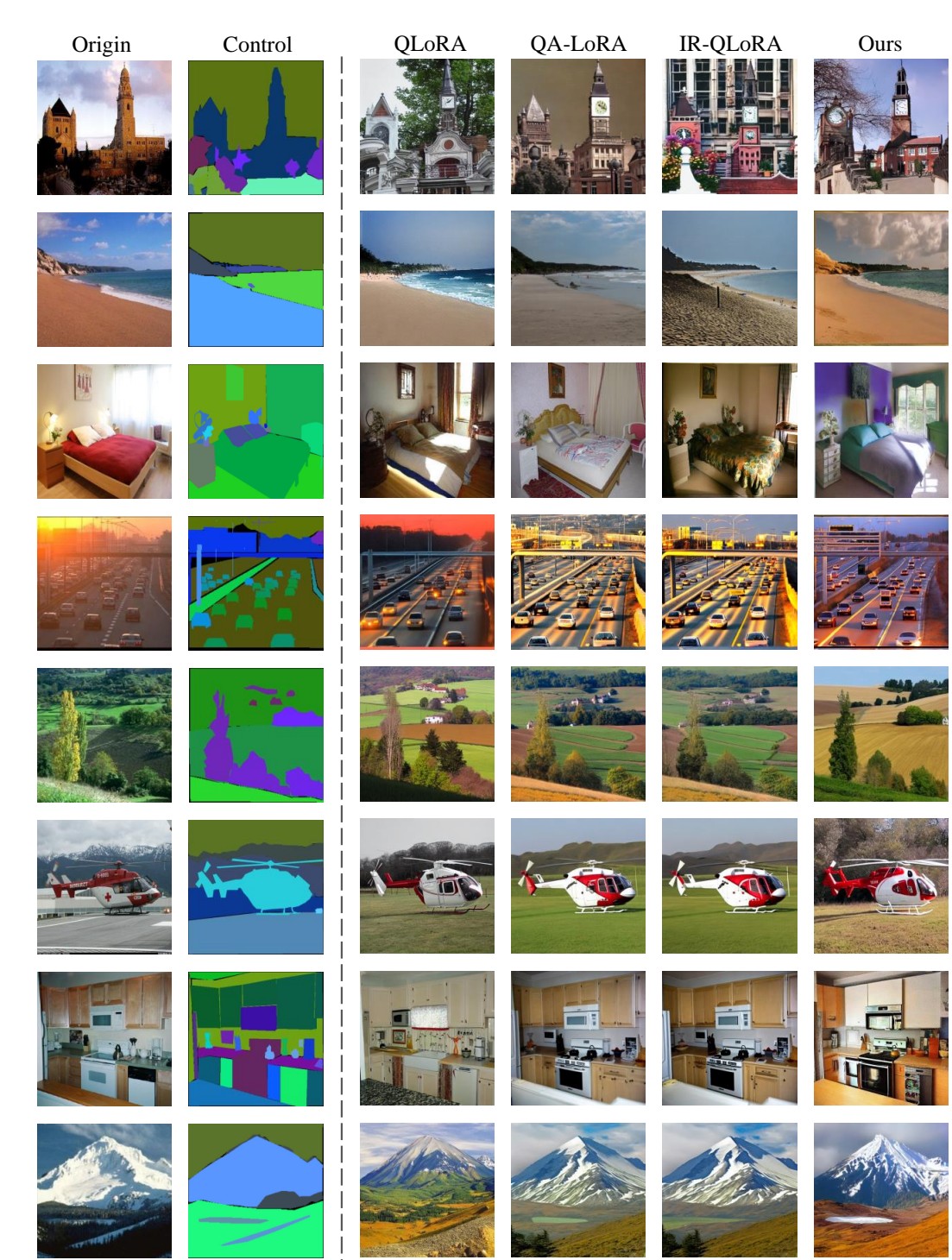

Figure 15: More qualitative comparison results on segmentation to image task. The 'Ours' denotes the IntLoRA$_{\text{SHIFT}}$. Zoom in for better effects.

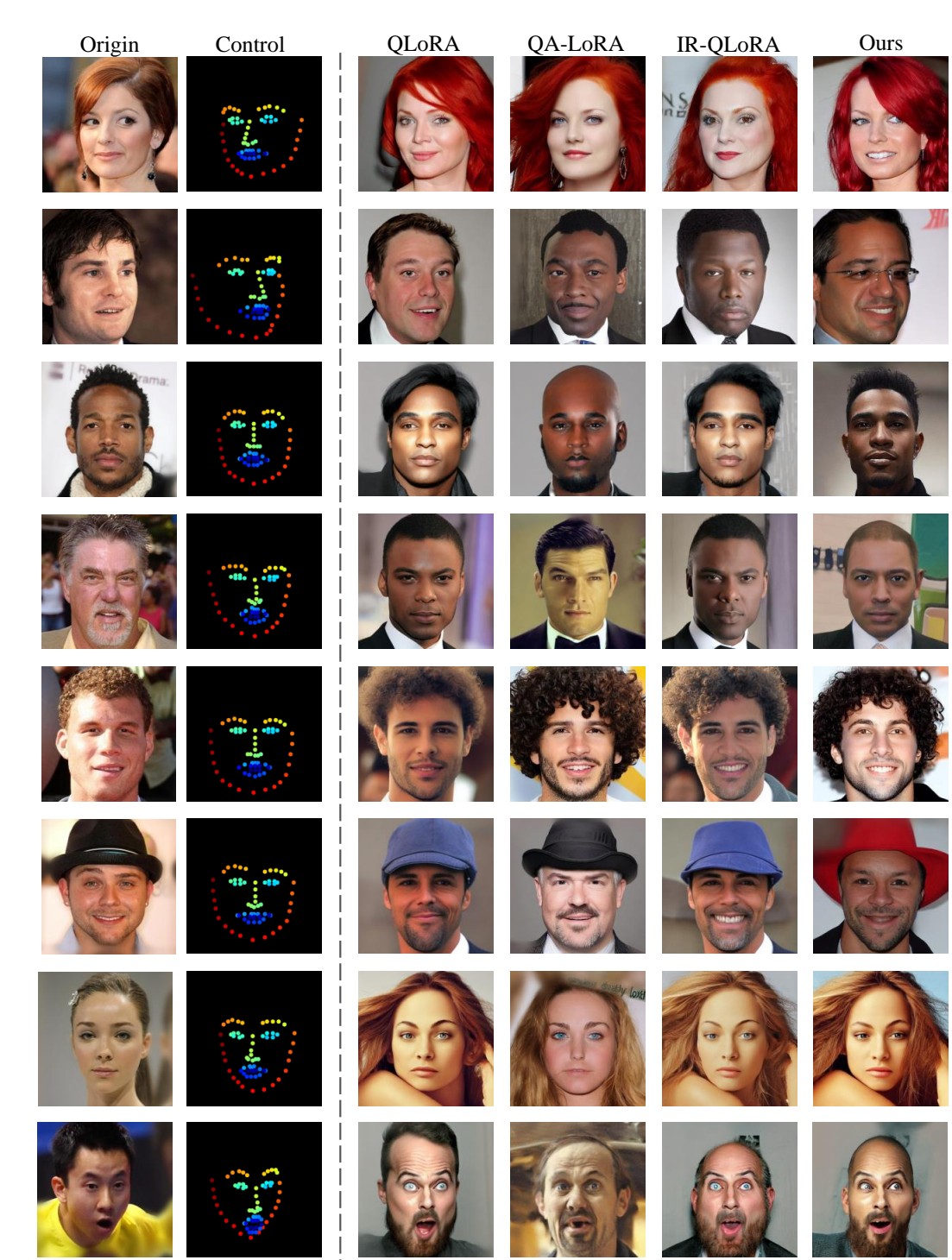

Figure 16: More qualitative comparison results on landmark to face task. The 'Ours' denotes the IntLoRA$_{\text{SHIFT}}$. Zoom in for better effects.

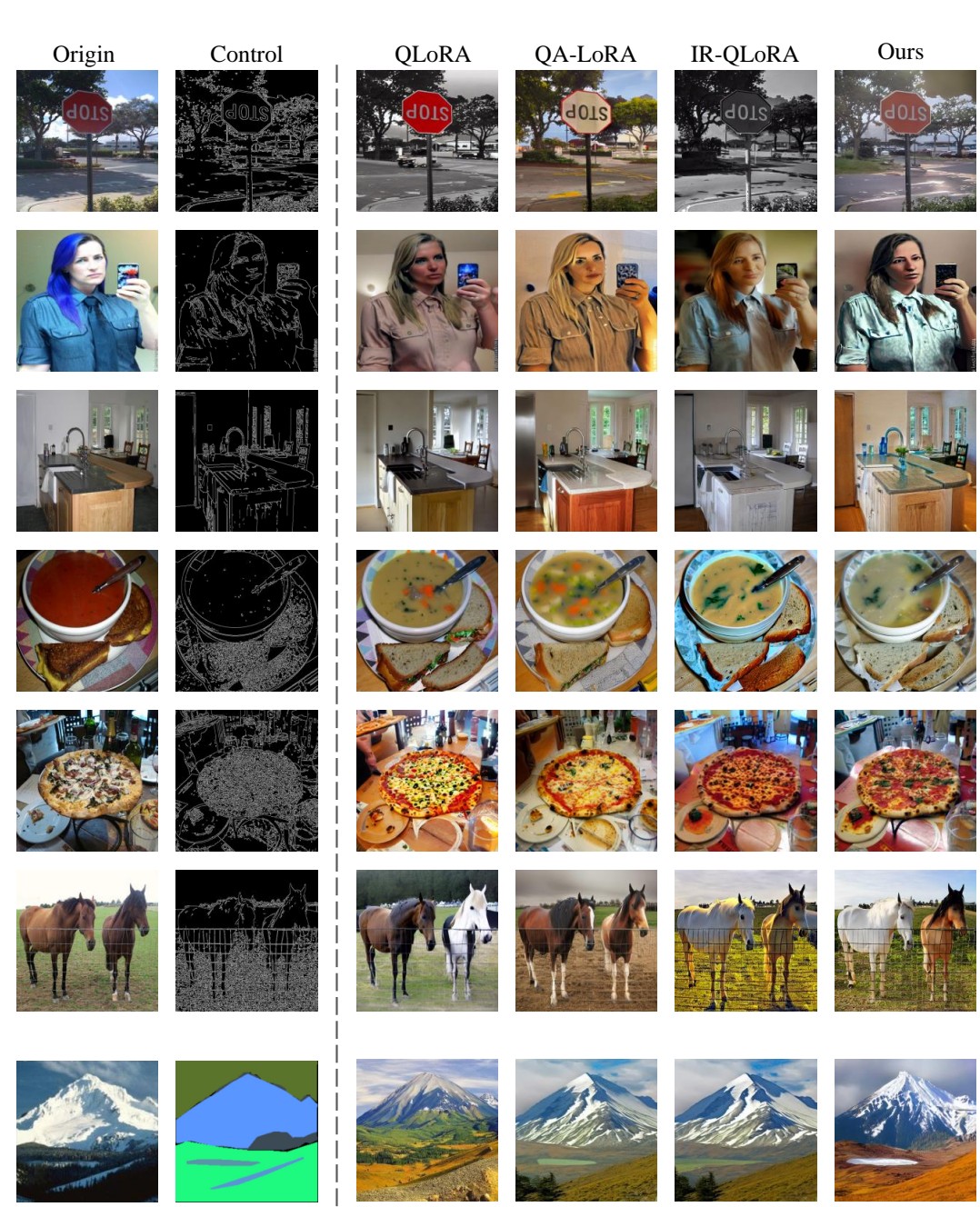

Figure 17: More qualitative comparison results on canny to image task. The 'Ours' denotes the IntLoRA$_{\text{SHIFT}}$. Zoom in for better effects.

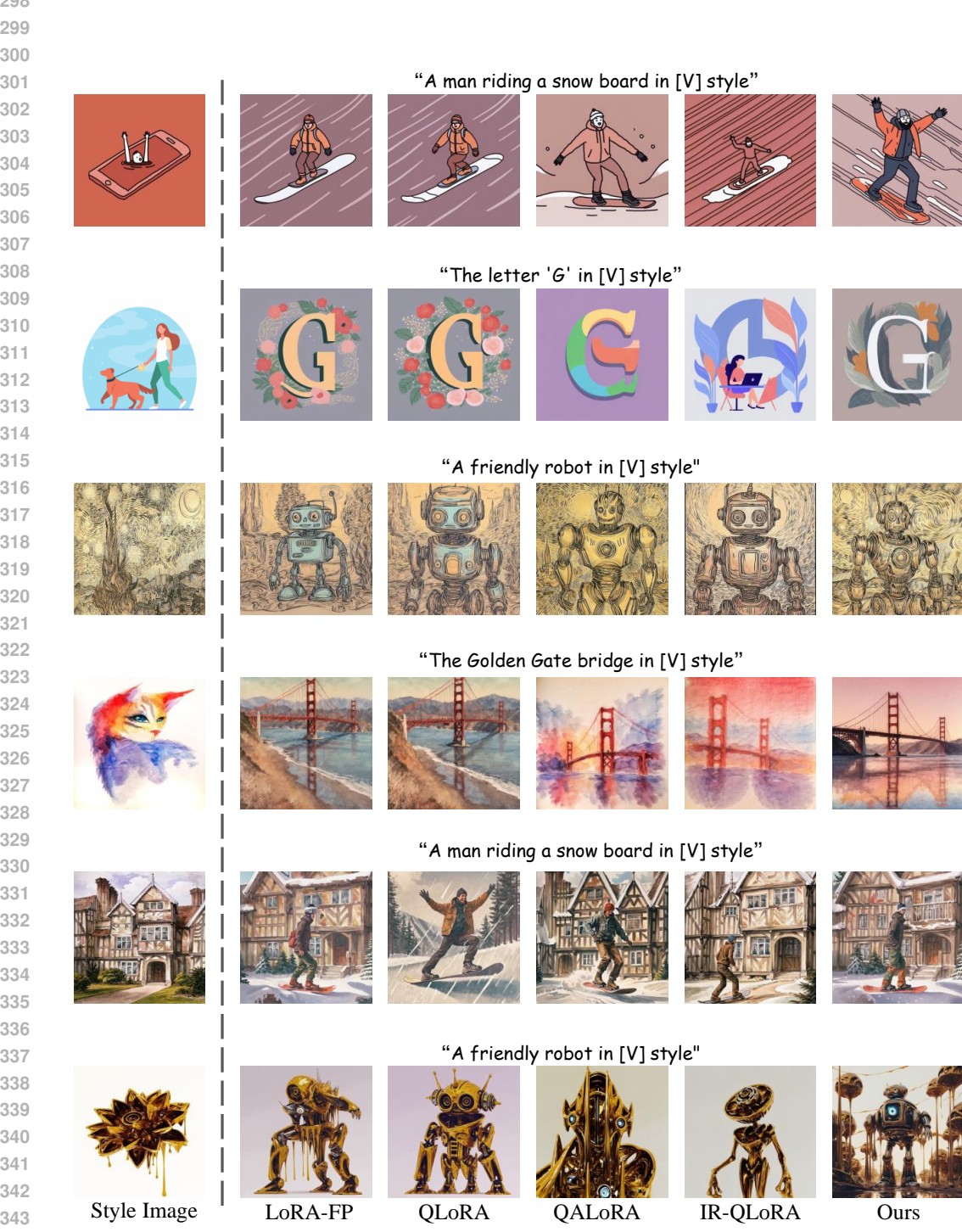

Figure 18: More qualitative comparison results on style-accustomed generation. The 'Ours' denotes the IntLoRA$_{\text{SHIFT}}$. Zoom in for better effects.

**Style images for training**

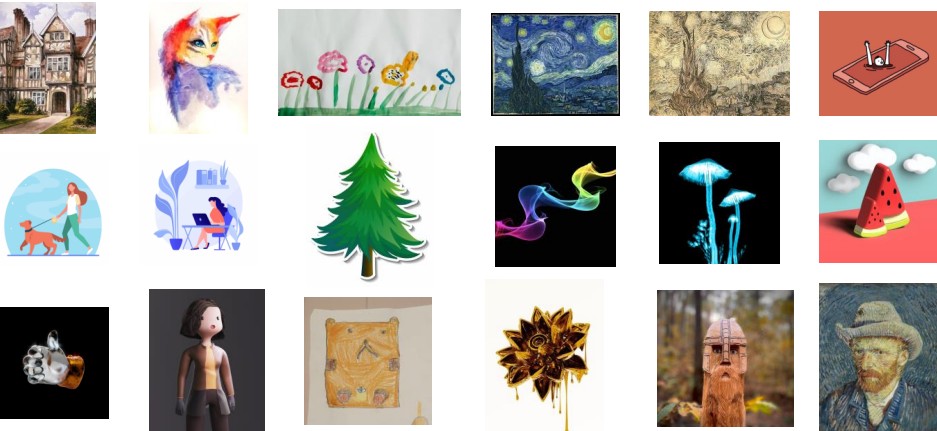

---

**Evaluation text prompt**

● "A friendly robot in [V] style",

● "A fluffy baby sloth with a knitted hat trying to figure out a lap top in [V] style"

● "A panda eating bamboo in [V]style"

● "The Golden Gate bridge in [V] style"

● "A man riding a snow board in [V] style"

● "The letter 'G' in [V] style"

Figure 19: The style images used for training and the text prompts for evaluation.

## O.1 CASES OF MATHEMATICAL QUESTION ANSWERING.

---

### Case Study 1

**Question**: Janet's ducks lay 16 eggs per day. She eats three for breakfast every morning and bakes muffins for her friends every day with four. She sells the remainder at the farmers' market daily for $2 per fresh duck egg. How much in dollars does she make every day at the farmers' market? **(Ground-truth answer is 18)**

**LoRA-FP Answer**: Janet's ducks lay 16 eggs per day. She eats three for breakfast every morning, so she has 16 - 3 = 13 eggs left. She bakes muffins for her friends every day with four eggs, so she has 13 - 4 = 9 eggs left. She sells the remainder at the farmers' market daily for $2 per fresh duck egg, so she makes 9 * $2 = $18. Therefore, Janet makes $18 every day at the farmers' market. The answer is 18.

**QLoRA Answer**: Janet's ducks lay 16 eggs per day. She eats three for breakfast every morning, so she has 16 - 3 = 13 eggs left. She bakes muffins for her friends every day with four eggs, so she has 13 - 4 = 9 eggs left. She sells the remainder at the farmers' market daily for $2 per fresh duck egg, so she makes 9 * $2 = $18. Therefore, Janet makes $18 every day at the farmers' market. The answer is: 18

**QA-LoRA Answer**: Janet's ducks lay 16 eggs per day. She eats three for breakfast every morning, so she has 16 - 3 = 13 eggs left. She bakes muffins for her friends every day with four, so she has 13 - 4 = 9 eggs left. She sells the remainder at the farmers' market daily for $2 per fresh duck egg, so she makes 9 * $2 = $18. Therefore, Janet makes $18 every day at the farmers' market. The answer is: 18

**QA-LoRA Answer**: Janet lays 16 eggs per day. She eats three for breakfast, so she has 16 - 3 = 13 eggs left. She bakes muffins with four eggs, so she has 13 - 4 = 9 eggs left. She sells the remainder at the farmers' market for $2 per egg, so she makes 9 * $2 = $18. Therefore, Janet makes $18 every day at the farmers' market. The answer is: 18

**Our Answer**: Janet lays 16 eggs per day. She eats three for breakfast, so she has 16 - 3 = 13 eggs left. She bakes muffins with four eggs, so she has 13 - 4 = 9 eggs left. She sells the remainder at the farmers' market for $2 per fresh duck egg, so she makes 9 * $2 = $18. Therefore, Janet makes $18 every day at the farmers' market. The answer is: 18.

---

## Case Study 2

**Question**: A robe takes 2 bolts of blue fiber and half that much white fiber. How many bolts in total does it take? **(Ground-truth answer is 3)**

**LoRA-FP Answer**: We know that the robe takes 2 bolts of blue fiber. To find the amount of white fiber, we can use the ratio of blue fiber to white fiber, which is 2:1. So, the amount of white fiber is half of the blue fiber, which is 2/2 = 1 bolt. Therefore, the total number of bolts needed is 2 + 1 = 3 bolts. The answer is 3.

**QLoRA Answer**: The robe takes 2 bolts of blue fiber. To find the amount of white fiber, we can use the ratio of 1:2. Since the ratio is 1:2, we can set up the equation 1/2 * 2 = x, where x represents the amount of white fiber. Solving for x, we get x = 1. Therefore, the robe takes 2 bolts of blue fiber and 1 bolt of white fiber. In total, the robe takes 2 + 1 = 3 bolts of fiber. The answer is: 3

**QA-LoRA Answer**:The robe takes 2 bolts of blue fiber. The robe takes half that much white fiber, which is 2/2 = 1 bolt of white fiber. So, in total, the robe takes 2 bolts of blue fiber + 1 bolt of white fiber = 3 bolts of fiber. Therefore, the robe takes 3 bolts of fiber in total. The answer is: 3

**IR-QLoRA Answer**: The robe takes 2 bolts of blue fiber. The robe takes half that much white fiber, which is 2/2 = 1 bolt of white fiber. So, the robe takes 2 bolts of blue fiber + 1 bolt of white fiber = 3 bolts in total. The answer is: 3

**Our Answer**: If a robe takes 2 bolts of blue fiber, then it takes 2/2 = 1 bolt of white fiber. So, the total number of bolts needed is 2 bolts of blue fiber + 1 bolt of white fiber = 3 bolts in total. The answer is: 3.

