# OpenReview forum: "IntLoRA: Integral Low-rank Adaptation of Quantized Diffusion Models"
_ICLR.cc/2025/Conference — Submitted to ICLR 2025_

### Official Review · Reviewer_c58Y · 2024-10-31

**Soundness:** 3
**Presentation:** 3
**Contribution:** 3
**Rating:** 6
**Confidence:** 4

**Summary:**

The paper introduces IntLoRA, which uses integral low-rank parameters to adapt quantized diffusion models. First, Adaptation-quantization separation (AQS) is proposed to overcome the difficulty of vanilla zero-initialized LoRA parameters is not quantization-friendly. Then, Multiplicative low-rank adaptation is proposed to enable the fusion of quantized weights and LoRA parameters. Variance matching control is further proposed to make the distribution suitable for log2-quantization. Experimental results show that IntLoRA improves the efficiency of adaptation while maintaining performance.

**Strengths:**

1.	Reducing storage costs is significant for adaptation tasks.
2.	The idea of multiplicative low-rank adaptation is novel.

**Weaknesses:**

1.	The training costs are not reported in the paper.
2.	In section 5.2, the authors mentioned that “IntLoRA is the only one that achieves INT type adaptation weights”, which is not true since EfficientDM also uses quantized LoRA weights.

**Questions:**

See weaknesses above.

---

> ### Author Response · Authors · 2024-11-21
> **Rebuttal by Authors**
>
> ## [Q1: Discussion on the training costs]
>
> > The training costs are not reported in the paper.
>
> As suggested, we compare on the training speed and GPU usage on the SD-1.5 based Dreambooth finetuning task. We use the NVIDIA 3090 GPU for testing, and the results are shown as follows.
>
> | setup             | training-speed | GPU-usage | Inference acceleration | Additional PTQ |
> |-------------------|----------------|-----------|-------------------------|----------------|
> | Full-finetune     | 0.74s/img      | 19.4G     | No                      | Yes            |
> | LoRA-FP           | 0.68s/img      | 6.7G      | No                      | Yes            |
> | QLoRA(W8-only)    | 0.71s/img      | 6.9G      | No                      | Yes            |
> | QLoRA(W8A8)       | 0.85s/img      | 7.2G      | Yes                     | Yes            |
> | IntLoRA(W8-only)  | 0.72s/img      | 7.1G      | No                      | No             |
> | IntLoRA(W8A8)     | 0.87s/img      | 7.4G      | Yes                     | No             |
>
>
> It can be seen that our IntloRA exhibits slightly larger training costs than LoRA-FP. This makes sense due to the fact that QAT typically consumes more cost than the common training counterpart. However, our IntLoRA can achieve **accelerated inference** benefiting from the quantization despite its slightly larger training cost than LoRA-FP. Moreover, comparing with QLoRA, which also uses QAT-like training, our IntloRA achieves similar training complexity, but our approach does not require additional PTQ after tuning, facilitating user-customized fast and low-cost deployment.
>
> ## [Q2: Contribution on INT type adapter]
>
> > In section 5.2, the authors mentioned that “IntLoRA is the only one that achieves INT type adaptation weights”, which is not true since EfficientDM also uses quantized LoRA weights.
>
> We would like to clarify that EfficientDM actually implements **quantized merged weights**, i.e., $\mathcal{Q}(\mathbf{W+AB})$, rather than our IntLoRA-like **quantized adapter weights**. This difference is also well recognized by Reviewer wQ3r.
>
> Quantized adapter weights are more challenging compared to quantized merged weights. This is because quantized merged weights only need to quantize the summation result of $\mathbf{W}$(FP16) and $\mathbf{AB}$(FP16), whereas quantized adapter weights require the quantization parameters of the adapters match those of the pre-trained weights for subsequent weight merging. Our IntloRA employs techniques including AQS, MLA and VMC to achieve the quantized adapter weights, thus facilitating seamless quantized weight merge without additional PTQ.

---

> > ### Comment · Reviewer_c58Y · 2024-11-22
> >
> > Thanks to the authors for the response. I will keep my score.

---

### Official Review · Reviewer_SB93 · 2024-11-01

**Soundness:** 3
**Presentation:** 3
**Contribution:** 3
**Rating:** 6
**Confidence:** 4

**Summary:**

The paper introduces IntLoRA, a pioneering method for adapting quantized diffusion models to optimize both training and inference efficiencies. It tackles the arithmetic inconsistency between low-rank parameters and pre-trained model weights by employing integer-based low-rank parameters. During training, the method applies quantization to the pre-trained model weights, and post-training, integrates these integer low-rank parameters seamlessly for efficient inference. To facilitate quantization-aware training, the authors propose the Adaptation-Quantization Separation (AQS), which allows for the coexistence of zero-initialized gradients and a quantization-friendly distribution. Additionally, a Variance Matching Control (VMC) mechanism is developed to finely tune the channel-aware variance, ensuring precise adaptation control.  Extensive experimental results indicate that IntLoRA not only competes with but often surpasses traditional methods in performance, significantly reducing memory usage and computational demands.

**Strengths:**

The paper demonstrates a strong and clear motivation by addressing a critical challenge in the domain of machine learning models—specifically, the adaptation of quantized diffusion models using integer low-rank parameters. This innovative approach effectively bridges the gap between maintaining precision and enhancing computational efficiency. Furthermore, the paper successfully resolves the prevalent issue of arithmetic inconsistency between pre-trained weights and adaptation parameters.

**Weaknesses:**

1.	The clarity of some explanations could be improved, particularly concerning why the all-zero distribution is not quantization-friendly. The text should elucidate why such a distribution necessitates a distinct quantizer design at the beginning of tuning. Providing a detailed example or analytical explanation on gradients would greatly enhance understanding, highlighting the specific challenges and potential impacts on the model’s performance during quantization.

2.	In the description of the VMC, the authors introduce a scalar $\alpha$ used as an exponent of $r$ to finely adjust the search for the optimal auxiliary matrix. However, the method for conducting this search remains unclear. It would be beneficial for the authors to specify the objective function used in this search process. Clarifying how $\alpha$ and $r$ are optimized, including the criteria and algorithms employed, would provide a more comprehensive understanding of the method’s effectiveness and implementation.

**Questions:**

1. The proposed method, IntLoRA, demonstrates substantial potential to enhance the performance of quantized models beyond simple adaptation. Integrating IntLoRA’s innovative techniques, such as integer low-rank parameters and network-wide reconstruction calibration, not only maintains but also significantly boosts the efficacy of models operating under quantization constraints.

2. The authors introduce the use of an auxiliary matrix in the AQS. It would be helpful to clarify whether the introduction of this auxiliary matrix introduces additional quantization difficulty like outliers to the pre-trained model weights.

3. The paper overlooks an important baseline in its experimental setup or discussion. The study “EfficientDM” by He et al., 2023, which similarly employs low-rank parameters for fine-tuning pre-trained models, is notably absent. Like IntLoRA, EfficientDM also allows for the merging of low-rank parameters into pre-trained model weights post-training. Including this baseline could enhance the comparative analysis, providing a clearer benchmark of IntLoRA’s performance against existing methodologies with similar strategies.

4. The authors merge integer low-rank parameters with quantized pre-trained model weights after training, a process that might lead to value overflow. It would be beneficial for the paper to address how this potential issue is managed. Specifically, detailing the mechanisms or techniques employed to prevent or mitigate overflow—such as scaling factors, normalization methods, or safeguards within the merging algorithm—would provide a clearer understanding of the reliability of the proposed method in handling data integrity during integration.

---

> ### Author Response · Authors · 2024-11-21
> **Rebuttal by Authors--Part1**
>
> ## [Q1: Why the all-zero distribution is not quantization-friendly]
>
> > The text should elucidate why such a distribution necessitates a distinct quantizer design at the beginning of tuning. Providing a detailed example or analytical explanation on gradients would greatly enhance understanding, highlighting the specific challenges and potential impacts on the model’s performance during quantization.
>
> Thank you for the comment. The all-zero distribution is difficult to quantize due to its limited value ranges. In detail, given the scaling factor $s = [\max(X)-\min(X)] / (2^b-1)$, this all-zero distribution leads to $min(X) = max(X)$, i.e. $s=0$.  Considering that the quantization of $X$ in Eq.1 appears the division by $s$, this all-zero distribution can lead to large quantization error due to the infinity value.
>
> We also attempt to explain this using the gradient analysis as you suggested. Specifically, since the AQS is proposed to avoid this all-zero distribution, we thus remove AQS from IntLoRA to obtain the zero-initilized LoRA weights and we try to train it to observe the gradients and performance. The results are as bellow.
>
>
> | Setup   | Gradient norm | DINO  |
> |---------|---------------|-------|
> | w/o AQS | nan           | 0.001 |
> | w/ AQS  | 0.42          | 0.381 |
>
>
> We observe that the training collapses at the beginning with a nan gradients when we remove the AQS to train a  zero-initilized variant. This ablation further validates the necessity of our AQS.
>
>
> ## [Q2: The Determination of $r$ and $\alpha$]
>
> > In the description of the VMC, the authors introduce a scalar $\alpha$ used as an exponent of $r$ to finely adjust the search for the optimal auxiliary matrix. However, the method for conducting this search remains unclear. It would be beneficial for the authors to specify the objective function used in this search process. Clarifying how $\alpha$ and $r$ are optimized, including the criteria and algorithms employed, would provide a more comprehensive understanding of the method’s effectiveness and implementation.
>
> In the proposed VMC, we use equation $r = \sigma_\mathbf{W}/\sigma_\mathbf{R}$ to determine the $r$, as illustrated in Line273 in the submission. After that, we use grid search from 0 to 2 with a step size 0.1 to determine the optimal $\alpha$. The objective function of the grid search is task-oriented, i.e., we use the task metrics to select the optimal $\alpha$, and the performance with varying $\alpha$ is shown in Fig.7.
>
> ## [Q3:  Integerate IntLoRA to Network-wide reconstruction]
>
> > The proposed method, IntLoRA, demonstrates substantial potential to enhance the performance of quantized models beyond simple adaptation. Integrating IntLoRA’s innovative techniques, such as integer low-rank parameters and network-wide reconstruction calibration, not only maintains but also significantly boosts the efficacy of models operating under quantization constraints.
>
> As suggested, we apply our IntLoRA to the diffusion model quantization, similar to the setup of EfficientDM. The experimental results on W4A4 quantized LDM-4 model on the ImageNet 256x256 image generation task are given in the global author response. The results show that our IntLoRA can also be applied to the network quantization, demonstrating the generality of the proposed method.
>
>
>
> ## [Q4: Impacts from the auxiliary matrix]
>
> > The authors introduce the use of an auxiliary matrix in the AQS. It would be helpful to clarify whether the introduction of this auxiliary matrix introduces additional quantization difficulty like outliers to the pre-trained model weights.
>
> In fact, the proposed VMC can effectively mitigate the outliers from the auxiliary matrix $\mathbf{R}$. Specifically, we use the variance ratio $r = \sigma_\mathbf{W}/\sigma_\mathbf{R}$ in the VMC to rescale the original sampled $\mathbf{R}$, which ensures that the value range of $\mathbf{R}$ is smaller than the one of $\mathbf{W}$, thus avoiding the introduction of additional outliers.
>
> To validate it, we also give the distribution of the $\mathbf{W}$, as well as the distribution of VMC-rescaled R in Fig.13 Appendix-G in the revised PDF. It can be seen that the range of $\mathbf{R}$ is well covered by $\mathbf{W}$, thus ensuring a minimal quantization error of $\mathcal{Q}(\mathbf{W-R})$.

---

> ### Author Response · Authors · 2024-11-21
> **Rebuttal by Authors--Part2**
>
> ## [Q5: Comparison with EfficientDM]
>
> > The paper overlooks an important baseline in its experimental setup or discussion. The study “EfficientDM” by He et al., 2023, which similarly employs low-rank parameters for fine-tuning pre-trained models, is notably absent. Like IntLoRA, EfficientDM also allows for the merging of low-rank parameters into pre-trained model weights post-training. Including this baseline could enhance the comparative analysis, providing a clearer benchmark of IntLoRA’s performance against existing methodologies with similar strategies.
>
> We provide a comprehensive comparison with EfficientDM in the global author rebuttal [G2], including using the quantized merged weights of EfficientDM for downstream fine-tuning tasks, as well as using our integer adapter for diffusion model quantization.
>
> From these experimental results, one can see that our IntLoRA can achieve better performance and even outperforms EffcientDM on W4A4 network quantization. It is worth noting that this work mainly focuses on downstream adaptation considering the benefits from the quantized adapter and seamless quantized merging. We will give a detailed discussion with EffciientDM in the revised version.
>
> ## [Q6: Strategy to overcome value overflow during merging]
>
> > The authors merge integer low-rank parameters with quantized pre-trained model weights after training, a process that might lead to value overflow. It would be beneficial for the paper to address how this potential issue is managed. Specifically, detailing the mechanisms or techniques employed to prevent or mitigate overflow—such as scaling factors, normalization methods, or safeguards within the merging algorithm—would provide a clearer understanding of the reliability of the proposed method in handling data integrity during integration.
>
> First, the proposed IntLoRA-SHIFT, which directly bit-shifts the INT4/INT8 weights as shown in Eq.9, does not face the numerical overflow problem, since the 1/2^N in the equation can act as a scaling factor. Therefore, we mainly discuss the machinsm to overcome value-overflow with IntLoRA-MUL of Eq.7. In detail, we can accumulate intermediate multiplication values in a higher precision and then we can scale and quantize the result back to INT4. For instance, during the multiplication between INT4 low-rank parameters and pre-trained weights,  we can use INT32 to hold the intermediate result. After that, we quantize the INT32 result as the product of the FP32 $s'$  and the INT4 merged weight. Finally, the FP32 $s'$ is multiplied with the original FP32 $s$ to obtain the final scaling factor and the INT4 merged weights.

---

> ### Author Response · Authors · 2024-11-23
> **A Kind Reminder of the Approaching Discussion Deadline**
>
> Dear Reviewer SB93,
>
> Thank you for taking the time and effort to review our paper. We have carefully addressed your valuable comments in detail.
> As the discussion phase is coming to an end soon, we kindly invite any further comments or suggestions you may have. If we have addressed your concerns, please kindly consider upgrading the score.
>
> We sincerely appreciate your efforts, which have significantly contributed to improving our manuscript.
>
> Best regards,
> The Authors

---

> > ### Comment · Reviewer_SB93 · 2024-11-23
> > **Thanks for the detailed response**
> >
> > Thank you for the detailed response and the additional experiment. I will keep my score.

---

### Official Review · Reviewer_qTL5 · 2024-11-03

**Soundness:** 3
**Presentation:** 2
**Contribution:** 3
**Rating:** 6
**Confidence:** 4

**Summary:**

This paper introduces IntLoRA, a method that quantizes the LoRA branch to integers for parameter-efficient fine-tuning (PEFT) in diffusion models. Traditional approaches typically use 16-bit precision for LoRA branches and focus on quantizing only the base model during fine-tuning. However, for inference, these methods often require re-quantization when the LoRA branches are fused with the base model. Unlike prior methods, such as QLoRA, IntLoRA fine-tunes the LoRA branches in a quantization-aware manner. To maintain adapter performance, it incorporates techniques like Adaptation-Quantization Separation (AQS), Multiplicative Low-rank Adaptation (MLA), and Variance Matching Control (VMC), and supports both linear and log2 quantization. Extensive experiments across various downstream tasks—including subject-driven generation, controllable generation, and style customization—demonstrate the effectiveness of IntLoRA in both visual quality and model size reduction.

**Strengths:**

* The paper presents a strong motivation, addressing a key limitation in current quantization methods: the need for re-quantization after applying adapters. This limitation has prevented the wide adoption of quantization methods in diffusion models.
* The concepts of Adaptation-Quantization Separation (AQS), Multiplicative Low-rank Adaptation (MLA), and Variance Matching Control (VMC) are innovative and well-supported by mathematical analysis. Their effectiveness is further validated through comprehensive experimental results.
* The authors' support for log2 quantization, a challenging but highly hardware-efficient setting that further enhances the practical utility of IntLoRA.

**Weaknesses:**

* The paper exclusively focuses on weight-only quantization, where activations remain at 16-bit precision. Since diffusion models are primarily compute-bound, quantizing only the weights does not yield speedups during inference. Additionally, in a weight-only quantization setting, users can choose not to fuse adapters, thereby avoiding re-quantization. As a result, the primary benefit of IntLoRA is just a reduction in adapter size compared to conventional PEFT methods—a relatively minor contribution, given that LoRA adapters are already small.
* he proposed PEFT method does not reduce training memory usage compared to existing PEFT methods (e.g., QLoRA). Furthermore, quantization-aware training of adapters may slow down training and even increase memory usage. The paper should discuss the training speed and memory usage.
* Clarification on storage requirements for $\mathbf{R}$ is necessary. Storing $\mathbf{R}$ after training the adapters would require significant storage, which may limit the method's practicality. If $\mathbf{R}$ is not stored, the authors should explain how it is obtained during inference. Also, note that $\mathbf{R}$ should derive from $\mathbf{W}_{\text{round}}$ rather than $\mathbf{W}$, as only quantized weights are accessible during PEFT.
* Writing:
  * Using "FP" to refer to original LoRA training is ambiguous, as there are low-precision formats like FP8 or FP4. Consider using "16-bit" instead. Note also that QLoRA uses NF4 for weight representation, which is not integer-based. Therefore, Lines 190–202 need to be rewritten.
  * Line 156: The dimension of $\mathbf{B}$ should be $d \times C_{\text{in}}$.
  * Figure 2: The caption lacks detail, and the figure is difficult to interpret.
  * Line 229: Refer to Section 5.3 on how to determine $\mathbf{R}$.
  * Line 265: Need to clarify $\rho$ is the correlation coefficient.
  * Table 4: Correct "FP addtion" to "FP addition".
  * Figure 7: Add axis labels for clarity.

**Questions:**

* Line 252: Could you provide any insights or justification for the assumption that $\mathbf{A}\mathbf{B}$ is orders of magnitude smaller than $\mathbf{R}$?
* Line 260: What is the purpose of a zero-mean adaptation term, and why is it better?
* Table 4: Why does the storage reduction exceed $8\times$ (e.g., $9.8/1.2 > 8$ and $0.34/0.04 > 8$)? If the model is stored in 32 bits, the reduction should be slightly less than $8\times$ due to the additional overhead from zero points and scaling factors. If stored in 16 bits (the preferred format), the reduction should not exceed $4\times$.
* Figure 6: Could you clarify the difference between VMC $\sigma_{\mathbf{R}}$ and the appropriate $\sigma_{\mathbf{R}}$? Additionally, how can one observe the large quantization error from the second image in the second row of the figure?

---

> ### Author Response · Authors · 2024-11-21
> **Rebuttal by Authors--Part1**
>
> ## [Q1: Clarification of benefits from weight-only quantization]
>
> > The paper exclusively focuses on weight-only quantization, where activations remain at 16-bit precision. Since diffusion models are primarily compute-bound, quantizing only the weights does not yield speedups during inference.
>
> Thanks for rising this concern. We would like to point out that the weight-only quantization of IntloRA can reduce GPU memory usage during fine-tuning, which is meaningful under the user customization senarios. Specifically, it is chanlleging for LoRA-FP tuning to load large diffusion models, e.g., just loading FLUX.1-dev[1] weights can cost 23.8GB GPU memory, on consumer-level GPUs. In contrast, our IntloRA can directly work on the quantized model, which reduces the loading GPU cost and facilitates diffusion model fine-tuning for user customization.
>
> [1] https://huggingface.co/black-forest-labs/FLUX.1-dev
>
>
> ## [Q2: IntLoRA with Activation Quantization]
>
> > Additionally, in a weight-only quantization setting, users can choose not to fuse adapters, thereby avoiding re-quantization. As a result, the primary benefit of IntLoRA is just a reduction in adapter size compared to conventional PEFT methods—a relatively minor contribution, given that LoRA adapters are already small.
>
> As suggested, we further supplement the **activation quantization** of IntLoRA. In detail, we employ the simple per-tensor Uniform Affine Quantization on the activation, without using other complicated tricks, and find it could work well on up to W4A8.  The results of Dreambooth fine-tuning are as follows, in which we use the average results of 15 text-subject pairs for evaluation.
>
> | methods         | nbits(W/A) | DINO   | CLIP-I  | CLIP-T  | LPIPS   |
> |-----------------|------------|--------|---------|---------|---------|
> | LoRA-FP        | W32A32     | 0.4828 | 0.6968  | 0.2954  | 0.8076  |
> | QA-LoRA        | W8A8       | 0.4156 | 0.6664  | 0.2834  | 0.8086  |
> | IR-QLoRA       | W8A8       | 0.4070 | 0.6630  | 0.2841  | 0.8110  |
> | IntLoRA-MUL    | W8A8       | 0.4498 | 0.6882  | 0.2858  | 0.8062  |
> | IntLoRA-SHIFT  | W8A8       | 0.4353 | 0.6842  | 0.2841  | 0.8257  |
> | QA-LoRA        | W4A8       | 0.4127 | 0.6897  | 0.2700  | 0.8281  |
> | IR-QLoRA       | W4A8       | 0.3722 | 0.6719  | 0.2707  | 0.8086  |
> | IntLoRA-MUL    | W4A8       | 0.4242 | 0.6913  | 0.2710  | 0.8181  |
> | IntLoRA-SHIFT  | W4A8       | 0.4039 | 0.6716  | 0.2709  | 0.8147  |
>
>
>
> It can be seen that our IntLoRA maintains superior performance under activation quantization settings. Importantly, the addition of activation quantization allows our IntLoRA to achieve practical **inference speed-up**.
>
>
> As for the W4A4 setup, we observe both our IntLoRA and all other baselines fail, which suggests additional techniques are needed for lower activation quantization bits. Considering that we mainly aim to address the challenge of arithmetic inconsistency during weight merge, we leave further 4-bit activation quantization for future work. Finally, we can also apply our IntLoRA to Diffusion Model Quantization, and even works well on the W4A4 setup.  Please refer to the global author rebuttal for related experimental results.

---

> ### Author Response · Authors · 2024-11-21
> **Rebuttal by Authors--Part2**
>
> ## [Q3: Discussion on training efficiency]
>
> > The proposed PEFT method does not reduce training memory usage compared to existing PEFT methods (e.g., QLoRA). Furthermore, quantization-aware training of adapters may slow down training and even increase memory usage. The paper should discuss the training speed and memory usage.
>
> The core goal of existing PEFT methods is to reduce the training cost, yet neglecting subsequent inference deployment. As recognized by Reviewer wQ3r and SB93, our method is **both** training and inference efficient. It is unfair to compare IntloRA to exiting PEFT methods **only** on the training end, since IntLoRA not only achieve QLoRA-like training efficiency, but also enables seamless quantized inference without additional PTQ.
>
> As  suggested, we compare on the training speed and GPU usage on the SD-1.5 based Dreambooth fine-tuning task. We use the NVIDIA 3090 GPU for testing, and the results are shown as follows.
>
>
> | setup             | traing-speed | GPU-usage | Inference acceleration | Additional PTQ |
> |-------------------|--------------|-----------|-------------------------|----------------|
> | Full-finetune     | 0.74s/img    | 19.4G     | No                      | Yes            |
> | LoRA-FP           | 0.68s/img    | 6.7G      | No                      | Yes            |
> | QLoRA(W8-only)    | 0.71s/img    | 6.9G      | No                      | Yes            |
> | QLoRA(W8A8)       | 0.85s/img    | 7.2G      | Yes                     | Yes            |
> | IntLoRA(W8-only)  | 0.72s/img    | 7.1G      | No                      | No             |
> | IntLoRA(W8A8)     | 0.87s/img    | 7.4G      | Yes                     | No             |
>
>
>
> It can be seen that our IntloRA exhibits slightly larger training costs than LoRA-FP. This makes sense due to the fact that QAT typically consumes more cost than the common training counterpart. However, the drawback of LoRA-FP is that it is difficult to speed up in the inference phase. Moreover, comparing with QLoRA, our IntloRA achieves similar training complexity, but our approach does not require additional PTQ, facilitating user-customized local deployment.
>
>
> ## [Q4: Storage requirements for $\mathbf{R}$]
>
> > Storing $\mathbf{R}$ after training the adapters would require significant storage, which may limit the method's practicality. If $\mathbf{R}$ is not stored, the authors should explain how it is obtained during inference. Also, note that $\mathbf{R}$ should derive from $\mathbf{W}_{round}$ rather than $\mathbf{W}$, as only quantized weights are accessible during PEFT.
>
> It should be noted that $\mathbf{R}$ is a distribution and is task-shared. Thus, given the distribution parameters and a fixed random seed, we can reproduce $\mathbf{R}$ online, meaning that $\mathbf{R}$ does not need to be stored.
>
> As for the derivation of $\mathbf{R}$, since we only need the max-value of $\mathbf{W}$ as mentioned in Line 409 of paper, we can thus approximate $max(\mathbf{W})$ as $s \cdot (max(\mathbf{W_{round}})-z)$, i.e., only $\mathbf{W}_{round}$ is needed to derive the VMC controlled $\mathbf{R}$.
>
> ## [Q5: About the writing]
>
> We will revise the paper based on the comments and update the PDF soon. Thank you for the detailed suggestions!

---

> ### Author Response · Authors · 2024-11-21
> **Rebuttal by Authors--Part3**
>
> ## [Q6:  Reply to other questions]
>
> > 6-1. Justification for $\mathbf{AB}$ being orders of magnitude smaller than $\mathbf{R}$
>
>
> We give the figure of the distribution of the learned $\mathbf{AB}$ and $\mathbf{R}$ in Fig.13 Appendix-H in the revised PDF. Since $\mathbf{AB}$ in LoRA is zero- initialized, it tends to be distributed around zero with learned small values aiming to not disturb the pre-training weights too much.
>
>
> > 6-2. What is the purpose of a zero-mean adaptation term, and why it is better?
>
> We prefer zero-mean adaptation term in order to facilitate subsequent log2 quantization on the adaptation term. This is due to the fact that most log2 buckets are around zero, e.g., $2^{-12}, 2^{-13}$, and thus a zero-mean adaptation term with the VMC controlled variance can force the values to near the zero, thus reducing log2 quantization error. We also provide the counter-example in Fig.6-Right, where the distribution of non-zero means results in very few log2 buckets being used, eventually leading to this setup cannot work.
>
>
>
> > 6-3. Table 4: Why does the storage reduction exceed 8× (e.g., 9.8/1.2>8 and 0.34/0.04>8)?
>
>
> This is due to the value round for presentation. For example, the original model is saved in FP32, and for 4-bit quantization settings, the reported results are obtained from $9.8/8 = 1.225 \approx 1.2$, and $0.34/8 = 0.0425 \approx 0.04$. We will make the values more precise in the revised version.
>
> > 6-4. Figure 6: Could you clarify the difference between VMC $\sigma_{R}$ and the appropriate $\sigma_{R}$ How can one observe the large quantization error from the second image in the second row of the figure?
>
> The VMC $\sigma_\mathbf{R}$ actually means the appropriate $\sigma_\mathbf{R}$, i.e., they convey the same concepts, and we will unify the term in the revision.
>
> The second image in the second row in Fig.6-Left is the reconstruction results of its left blue counterpart using the reconstruction equation: $W \approx \mathcal{Q}(W-\mathbf{R}) + \mathbf{R}$.  The difference between these two blue KDE curves indicates large reconstruction errors due to a too large $\sigma_\mathbf{R}$.

---

> ### Author Response · Authors · 2024-11-23
> **A Kind Reminder of the Approaching Discussion Deadline**
>
> Dear Reviewer qTL5,
>
> Thank you for taking the time and effort to review our paper. We have carefully addressed your valuable comments in detail.
> As the discussion phase is coming to an end soon, we kindly invite any further comments or suggestions you may have. If we have addressed your concerns, please kindly consider upgrading the score.
>
> We sincerely appreciate your efforts, which have significantly contributed to improving our manuscript.
>
> Best regards,
> The Authors

---

> > ### Comment · Reviewer_qTL5 · 2024-11-24
> >
> > Thanks for the author response. With the quantized activation results, my primary concern has been addressed. I still recommend that the authors further refine their writing. For instance, the caption of Figure 2 could be made more self-contained, and the detailed sample distribution of $\mathbf{R}$ should be formulated explicitly, clarifying that it can be restored from the distribution using a fixed seed. Additionally, in Figure 6, an arrow should be added from "large quantization" to the curve in the first figure of the second row to help readers understand that the differences between the curves indicate a significant error. I will raise my score to 6.

---

> > > ### Author Response · Authors · 2024-11-24
> > > **Official Responds by Authors**
> > >
> > > Dear Reviewer qTL5,
> > >
> > > Thank you very much for your positive feedback. We are delighted that our responses have addressed your concerns.
> > >
> > > We will add activation quantization results in the updated version. We will also further polish the paper based on the Reviewer qTL5' comments, e.g., to make Fig.2 and Fig.6 more understandable, to explicitly point out how $\mathbf{R}$ is generated, etc.
> > >
> > > Thanks again for your efforts and these helpful suggestions!
> > >
> > > Best regards, The Authors

---

### Official Review · Reviewer_wQ3r · 2024-11-05

**Soundness:** 3
**Presentation:** 3
**Contribution:** 2
**Rating:** 5
**Confidence:** 4

**Summary:**

The authors propose IntLoRA, which employes INT low-rank parameters to adapt the quantized diffusion models, to address the arithmetic inconsistency between the quantized pre-trained weights and adaptation weights. All the process will operate directly on INT arithmetic in training and merged weights at inference stage will also be on INT format. Apparently, it will speed up both the training and inference stage.

**Strengths:**

1) The presentation of challenges and contributions are clear to me.
2) The presentation of methods and experiments are clear to me.

**Weaknesses:**

1) Recent works, e.g., EfficientDM and LoftQ, have worked out quantized diffusion models using LoRA. Different from this paper, EfficientDM focus on quantized merged weights at inference stage. Can you compare with it? BTW, I am doubting the importance of the speed-up for the training stage as the pretrained weights is fixed, only small adapters are updated.  Can you show how much speed-up as compared to full-precision? or at least theorotically show the improvement?

2) From my understanding, the proposed methods also can be used in NLP tasks. As the baselines have results on NLP tasks in their papers, why not verify the proposed methods on NLP tasks? Could you show some experimental results on NLP tasks?

3) For diffusion models, seems it have standard/common verification on some tasks, please check EfficientDM paper. As all baselines are implemented by authors, I just doubt the reliablity because of hyper-parameter tuning. Maybe it is better compare with SOTA results in paper.

**Questions:**

Have asked in Weakness section.

---

> ### Author Response · Authors · 2024-11-21
> **Rebuttal by Authors**
>
> ## [Q1: Comparison with EfficientDM]
>
> > Recent works have worked out quantized diffusion models using LoRA. Different from this paper, EfficientDM focuses on quantized merged weights at inference stage. Can you compare with it?
>
> Thanks for your comment, we provide a detailed comparison with EfficientDM in the global rebuttal [G2].
>
> In short, our IntloRA is more suitable for accelerated downstream fine-tuning compared to EfficientDM. In addition, our approach can also potentially be used for diffusion model quantization and have shown promising results.
>
> ## [Q2: The significance of accelerated training]
>
> > I am doubting the importance of the speed-up for the training stage as the pretrained weights is fixed, only small adapters are updated. Can you show how much speed-up as compared to full-precision? or at least theoretically show improvement?
>
> Thank you for raising this concern. We would like to point out that LoRA only reduces the GPU memory of **gradient** and **optimizer states** during training, and cannot benefit the model **weights**. Considering that the diffusion models are getting bigger, e.g., just importing FLUX.1-dev to cuda can take up 23.8GB, it makes sense to reduce the size of the weights during training.
>
> Our IntloRA only needs the quantized pre-training weights during training, further facilitates user-customized diffusion fine-tuning on consumer-level GPUs. Theoretically, taking SDXL as an example, full fine-tuning SDXL  consumes 59GB GPU memory, of which 13.5GB comes from the weights. Using FP32 LoRA tuning consumes 28.38GB, indicating 30.62GB reduced gradients and optimizer states. However, this LoRA tuning still maintains the weights in FP32. Theoretically, our IntLoRA can further reduce this 13.5GB FP32 weight to 3.375GB with the INT8 quantization.
>
> Finally, we would like to remind that efficient training is only one of the benefits of our IntLoRA, whose another advantage is to obtain downstream quantized models without additional PTQ for efficient inference.
>
> ## [Q3: Results on NLP tasks]
>
> > From my understanding, the proposed methods also can be used in NLP tasks. As the baselines have results on NLP tasks in their papers, why not verify the proposed methods on NLP tasks? Could you show some experimental results on NLP tasks?
>
>
> We understand your concerns, that is why we conduct experiments on mathQA task with Llama3-8B model in Tab.6 in the Appendix-A, and we also give this result as follows. It can be seen that our W8 IntloRA fine-tuned model outperforms other quantization baselines, indicating the generalizability of the proposed method for other tasks.
>
> | method         | LoRA-FP | QLoRA  | QA-LoRA | IR-QLoRA | IntLoRA-MUL | IntLoRA-MUL |
> |----------------|---------|--------|---------|----------|-------------|-------------|
> | QA-accuracy    | 62.24%  | 64.06% | 63.53%  | 63.00%   | 64.32%      | 64.10%      |
>
>
> ## [Q4: Reliability of the results]
>
> > For diffusion models, seems it have standard/common verification on some tasks, please check EfficientDM paper. As all baselines are implemented by authors, I just doubt the reliability because of hyper-parameter tuning. Maybe it is better compare with SOTA results in paper.
>
> As suggested, we replace the EfficientDM-Layer(line 385) in `quant_layer.py` [5], in EfficientDM github repo with our IntLoRA-Layer and keep the others intact to validate on the diffision model quantization task. The results are shown in the global rebuttal [G2], and it can be seen our IntLoRA can achieve better performance.
> As discussed in the global author rebuttal [G2], our IntLoRA is more suitable for applying to downstream adaptation. To this end, in the paper, we follow the evaluation protocol in diffusion model fine-tuning works, including OFT[1], COFT[2], as our core experimental task setup.
>
> As for the reliability of the compared baselines, the implemented methods for comaprasion also follows their official code: QLoRA[3], QALoRA[4], IR-QLoRA[5], and we keep all other hyper-parameters the same and only change the module class of Quantized-Linear-Layer with different methods.
>
> Finally, in the attached suppl-zip, we also provide the code of our IntLoRA on Dreanbooth fine-tuning task. And we will open-souce all related code upon accptance.
>
> [1] Controlling Text-to-Image Diffusion by Orthogonal Finetuning. NeurIPS23. Code: https://github.com/Zeju1997/oft
>
> [2]Parameter-Efficient Orthogonal Finetuning via Butterfly Factorization. ICLR24. Code:https://github.com/wy1iu/butterfly-oft
>
> [3]QLoRA: https://github.com/artidoro/qlora
>
> [4]QALoRA: https://github.com/yuhuixu1993/qa-lora/blob/main/peft_utils.py
>
> [5]IR-QLoRA: https://github.com/htqin/IR-QLoRA/blob/main/integer%20quantizer/quantizer_icq.py
>
> [6]EfficentDM: https://github.com/ThisisBillhe/EfficientDM/blob/main/quant_scripts/quant_layer.py

---

> ### Author Response · Authors · 2024-11-23
> **A Kind Reminder of the Approaching Discussion Deadline**
>
> Dear Reviewer wQ3r,
>
> Thank you for taking the time and effort to review our paper. We have carefully addressed your valuable comments in detail.
> As the discussion phase is coming to an end soon, we kindly invite any further comments or suggestions you may have. If we have addressed your concerns, please kindly consider upgrading the score.
>
> We sincerely appreciate your efforts, which have significantly contributed to improving our manuscript.
>
> Best regards,
> The Authors

---

> ### Author Response · Authors · 2024-11-27
> **The Second Reminder of the Approaching Discussion Deadline**
>
> Dear Reviewer wQ3r,
>
> We notice it has been 6 days since our Author Rebuttal to your raised concerns. We understand your tight schedule, but given that the discussion phase is about to end, we sincerely hope that you can update your assessment if our response addresses your questions. If you have any other questions, we will actively discuss with you as well.
>
> Thank you for taking the time and effort to review our paper.
>
> Best regards, The Authors

---

> > ### Comment · Reviewer_wQ3r · 2024-12-02
> > **Extra concerns**
> >
> > Thank you for the detailed response. However, I still have several concerns:
> > 1. Compared to EfficientDM, if we consider only the inference stage, both EfficientDM and IntLoRA have the same target to obtain the well-quantized merged weights. Besides, I don't quite understand why IntLoRA can do downstream tasks and EfficientDM cannot in theory. To me, IntLoRA's main contribution is to save on training costs.  Thus, in the abstract and introduction, other contributions seem not very strong to me. Seems the authors didn't see EfficientDM paper before submission.
> >
> > 2. For experimental reproduction, we appreciate the authors for releasing their code. However, for a fair comparison, it might be more effective to conduct experiments on tasks already addressed by other papers. This would allow reviewers to more clearly assess the differences. Currently, the experiments, code, and setup are entirely designed by the authors even though the authors say they use public code. While this approach is commendable, it raises the question: why not use existing benchmarks and demonstrate that your method outperforms others, rather than focusing solely on new datasets and tasks? This feedback applies to experiments on diffusion model tasks and NLP tasks. I welcome any discussion.
> >
> > Thanks for the efforts in the rebuttal.

---

> > > ### Author Response · Authors · 2024-12-02
> > > **Author Responds to Extra Concerns**
> > >
> > > We thank the reviewers for being able to engage in the discussion and raise additional concerns before the closing discussion phase.
> > >
> > > (i) For the advantages of IntloRA over EfficentDM. Theoretically, EffcientDM could indeed be used for inference-accelerated downstream adaptation similar to our IntLoRA, but comes with a greater cost. First, EfficentDM needs to store quantized large models for EACH downstream task. Thus, **even if the merged models are quantized, storing multiple large quantized downstream weights still requires a huge overhead**. In contrast, our IntLoRA only needs to store quantized low-rank weights for each downstream task. Second, if we use EffcientDM's technique for downstream adaptation, it should be noted that **EffcientDM requires additional LoRA finetuning on the calibration dataset**, which is unacceptable for plug-and-play user-customized diffusion model scenarios.
> > > Given the above factors, we say that IntloRA is more suitable for downstream task adaptation than EfficientDM.
> > > Furthermore, the reviewer's “Seems the authors didn't see EfficientDM paper before submission.” is inaccurate. In fact, **we have actually cited EfficientDM paper in the Related Work section in the original submission version**. We have also provided a more detailed experimental discussion in the revised PDF during rebuttal.
> > >
> > >
> > > (ii) As discussed in (i), our IntLoRA is better suited for downstream adaptation, that is why we mainly focus on the downstream adaptation task in this work. Moreover, the reviewer's statement of "the experiments, code, and setup are entirely designed by the authors" looks factually incorrect since we use the standard adaptation setup of [1] and [2]. At last, following the Reviewer wQ3r's comment, **we also performed the same diffusion model quantization experiment similar to EfficietDM** in our global author rebuttal, and it seems the Reviewer wQ3r has ignored our efforts at this point.
> > >
> > >
> > > We welcome further active discussion and hope to be able to address your questions. Thanks for your response.
> > >
> > > Best regards, The Authors.
> > >
> > > [1] Controlling Text-to-Image Diffusion by Orthogonal Finetuning. NeurIPS23. Code: https://github.com/Zeju1997/oft
> > >
> > > [2] Parameter-Efficient Orthogonal Finetuning via Butterfly Factorization. ICLR24. Code:https://github.com/wy1iu/butterfly-oft

---

> > > > ### Author Response · Authors · 2024-12-03
> > > > **The Third Reminder of the Approaching Discussion Deadline**
> > > >
> > > > Dear Reviewer wQ3r,
> > > >
> > > > As you said, you are "welcome to any discussion". We responded to your extra concerns within one hour, but we haven't obtain the reply from you after 24h. Considering that the discussion phase will close in a couple of hours, we hope that you will respond positively.
> > > >
> > > > We believe that constructive and timely communication between reviewers and authors is essential for the benefit of both parties, and we are very eager to know if our response addressed your concerns as it is important to improve our work.
> > > >
> > > > Best regards, The Authors

---

### Author Response · Authors · 2024-11-21
**Global Author Rebuttal**

## [G1. Remarks by authors]

We would like to express our sincere gratitude to all the reviewers for taking their time reviewing our work and providing fruitful reviews that have definitely improved the paper. We are encouraged that they find our method

- "a strong and clear motivation " (Reviewer wQ3r, qTL5, SB93)
- "addressing a key limitation in current quantization methods" (Reviewer SB93 qTL5)
- "innovative and well-supported by mathematical analysis" (Reviewer qTL5, c58Y)
- "comprehensive experimental results"(Reviewer qTL5)

If you have any further questions, we will actively discuss with you during the author-reviewer discussion period. Due to limited time, the PDF has not been fully revised based on the reviewers' comments, we will upload the fully revised version in a few days.

## [G2. Detailed comparison with EfficientDM]

Despite both appear "diffusion" and "quantization", our IntLoRA focuses on the **quantization-aware fine-tuning**, instead of the **diffusion model quantization** of EfficientDM.  Importantly, **EfficientDM is more suitable for diffusion modeling quantization**, since directly quantizing the merged weights is sufficient given quantization aims to obtain **one** quantized model. For downstream task adaptation, where **multiple** quantized downstream models are required, the proposed **IntLoRA is more suitable for downstream adaptation**, which allows separately quantized weights and adapters as well as seamless weight merge.

However, considering Reviewer wQ3r, SB93, and c58Y's kind suggestions, we compare our IntLoRA in detail with EffiicentDM as follows.


### [G2-1: Limitations of EffieicntDM for downstream adaptation]

EfficientDM focuses on the quantized merged weights, i.e., $\mathcal{Q}(\mathbf{W+AB})$. An obvious drawback
is that it still needs to load the large FP32/FP16 weights during downstream adaptation, which is not suitable for user customization on consumer-level GPUs. For example, just loading the FP32 FLUX.1-dev can consumes over 28GB GPU memory. Moreover, $N$ downstream task need to store $N$ large models in EfficientDM, which also leads to noticeable challenge for customized model sharing among users like Civitai. As a comparison, our IntLoRA only need to load the quantized INT4/INT8 model during tuning,  and store 1 quantized pre-training weight & $N$ quantized lightweight LoRA weights to reduce storage cost.


### [G2-2: Applying EffiicentDM to downstream adaptation]

We also apply the EfficientDM technique to the downstream adaptation task of Dreambooth fine-tuning. We average the results of 15 text-subject pairs for evaluation. The results are shown below.

| setup          | nbits | DINO   | CLIP-I  | CLIP-T  | LPIPS   |
|----------------|-------|--------|---------|---------|---------|
| LoRA-FP       | W32   | 0.4828 | 0.6968  | 0.2954  | 0.8076  |
| EfficientDM   | W8    | 0.4055 | 0.6649  | 0.2855  | 0.8123  |
| IntLoRA-MUL   | W8    | 0.4094 | 0.6630  | 0.2857  | 0.8048  |
| EfficientDM   | W4    | 0.3733 | 0.6555  | 0.2847  | 0.8085  |
| IntLoRA-MUL   | W4    | 0.3749 | 0.6599  | 0.2867  | 0.8011  |

It can be seen that our IntLoRA-MUL achieves better performance than EfficientDM. Moreover, IntLoRA also enjoys benefits described in the above [G2-1]

### [G2-3: Applying IntLoRA to diffusion model quantization]

Since our IntLoRA can potentially be used in diffusion model quantization as well, we also apply our IntLoRA to the quantization experimental settings of EfficientDM using its official code. In detail, we use the LDM-4 on ImageNet256$\times$256 image generation tasks, and train the ddim_step=20 model with 500 training epochs. The results are as follows. We keep other hyper-parameters the same except replacing the original EfficientDM LoRA layer with our IntLoRA layer.

| methods         | nbits(W/A) | IS      | FID   | sFID   | precision | recall |
|-----------------|------------|---------|-------|--------|-----------|--------|
| EfficientDM    | W4A4       | 178.20  | 13.42 | 26.67  | 0.70      | 0.58   |
| IntLoRA-SHIFT  | W4A4       | 116.50  | 20.20 | 26.79  | 0.63      | 0.58   |
| IntLoRA-MUL    | W4A4       | 199.20  | 10.43 | 24.02  | 0.79      | 0.55   |


It can be seen that our IntLoRA-MUL achieves even better performance than the EfficientDM on weight-bits-4&activation-bits-4 setups.

---

### Meta-Review · Area_Chair_Xu2i · 2024-12-19

**Metareview:**

The paper titled "IntLoRA: Integral Low-rank Adaptation of Quantized Diffusion Models" introduces IntLoRA, a novel parameter-efficient fine-tuning (PEFT) method tailored for quantized diffusion models. The primary innovation lies in utilizing integer-based low-rank adaptation parameters, enabling seamless integration with quantized pre-trained weights through integer multiplication or bit-shifting. This approach aims to enhance both training and inference efficiencies by reducing memory usage, storage costs, and eliminating the need for additional post-training quantization (PTQ). The authors present comprehensive experimental results demonstrating that IntLoRA achieves performance comparable to or surpassing traditional LoRA methods while offering significant efficiency improvements across various downstream tasks, including computer vision and natural language processing (NLP).

Despite the promising contributions, several critical concerns undermine the paper's suitability for acceptance. Firstly, the comparison with existing methods such as EfficientDM and LoftQ is insufficiently addressed, particularly regarding the applicability and performance on standard benchmarks. Reviewer wQ3r highlighted the lack of comparisons on established NLP tasks and questioned the reliability of the experimental setups, suggesting potential biases due to hyper-parameter tuning. Although the authors provided additional experiments and clarifications during the rebuttal, the persistent concerns from Reviewer wQ3r regarding the method's novelty and practical significance remain unresolved. Additionally, the paper's focus on weight-only quantization limits the claimed inference speedups, as activation quantization is not comprehensively explored. These weaknesses, coupled with the unresolved doubts about the method's generalizability and comparative advantage, lead to the recommendation to reject the paper in its current form.

**Additional Comments On Reviewer Discussion:**

During the rebuttal phase, the authors addressed several key concerns raised by the reviewers. Reviewer wQ3r questioned the comparison with EfficientDM and the significance of training speedups, as well as the absence of experiments on NLP tasks. The authors provided detailed comparisons with EfficientDM, demonstrating that IntLoRA outperforms EfficientDM in certain settings and clarified the advantages in downstream task adaptation. They also included additional experimental results on the MathQA task using the Llama3-8B model, showcasing IntLoRA's applicability to NLP tasks. However, despite these responses, Reviewer wQ3r remained unconvinced, emphasizing the need for comparisons on standard benchmarks and expressing concerns about the method's practical impact.

Reviewer qTL5 initially leaned towards acceptance but raised concerns about weight-only quantization and training efficiency. The authors supplemented their submission with activation quantization results, addressing the primary concern and refining their explanations based on the reviewer's feedback. Consequently, Reviewer qTL5 upgraded their score to reflect the satisfactory resolution of their initial concerns. Reviewer SB93 also expressed a positive stance towards acceptance, acknowledging the authors' efforts in addressing feedback. However, the lingering reservations from Reviewer wQ3r, particularly regarding the method's comparative advantage and experimental rigor, were decisive factors leading to the overall rejection recommendation.

---

### Decision · Program_Chairs · 2025-01-22

Reject